# Changing the Training Data Distribution to Reduce Simplicity Bias Improves In-distribution Generalization

**Tuan Hai Dang Nguyen**   **Paymon Haddad**   **Eric Gan**   **Baharan Mirzasoleiman**
Department of Computer Science, UCLA

## Abstract

Can we modify the training data distribution to encourage the underlying optimization method toward finding solutions with superior generalization performance on *in-distribution* data? In this work, we approach this question for the first time by comparing the inductive bias of gradient descent (GD) with that of sharpness-aware minimization (SAM). By studying a two-layer CNN, we rigorously prove that SAM learns different features more uniformly, particularly in early epochs. That is, SAM is less susceptible to simplicity bias compared to GD. We also show that examples containing features that are learned early are separable from the rest based on the model's output. Based on this observation, we propose a method that (i) clusters examples based on the network output early in training, (ii) identifies a cluster of examples with similar network output, and (iii) upsamples the rest of examples only once to alleviate the simplicity bias. We show empirically that USEFUL effectively improves the generalization performance on the *original* data distribution when training with various gradient methods, including (S)GD and SAM. Notably, we demonstrate that our method can be combined with SAM variants and existing data augmentation strategies to achieve, to the best of our knowledge, state-of-the-art performance for training ResNet18 on CIFAR10, STL10, CINIC10, Tiny-ImageNet; ResNet34 on CIFAR100; and VGG19 and DenseNet121 on CIFAR10.

## 1   Introduction

Training data is a key component of machine learning pipelines and directly impacts its performance. Over the last decade, there has been a large body of efforts concerned with improving learning from a given training dataset by designing more effective optimization methods [22, 38, 76] or neural networks with improved structures [47, 56, 82] or higher-capacity [49, 51]. More recently, improving the quality of the training data has emerged as a popular avenue to improve generalization performance. Interestingly, higher-quality data can further improve the performance when larger models and better optimization methods are unable to do so [23, 27]. Recent efforts to improve the data quality have mainly focused on filtering irrelevant, noisy, or harmful examples [23, 45, 66]. Nevertheless, it remains an open question if one can change the distribution of a *clean* training data to further improve the *in-distribution* generalization performance of models trained on it.

At first glance, the above question may seem unnatural, as it disputes a fundamental assumption that training and test data should come from the same distribution [29]. Under this assumption, minimizing the training loss generalizes well on the test data [7]. Nevertheless, for overparameterized neural networks with more parameters than training data, there are many zero training error solutions, all global minima of the training objective, with *different generalization* performance [25]. Thus, one may still hope to carefully change the data distribution to drive the optimization algorithms towards finding more generalizable solutions on the *original* data distribution.

38th Conference on Neural Information Processing Systems (NeurIPS 2024).

In this work, we take the first steps towards addressing the above problem. To do so, we rely on recent results in non-convex optimization, showing the superior generalization performance of sharpness-aware-minimization (SAM) [22] over (stochastic) gradient descent (GD). SAM finds flatter local minima by simultaneously minimizing the loss value and loss sharpness. In doing so, it outperforms (S)GD and obtains state-of-the-art performance, at the expense of doubling the training time [20, 81]. Our key idea is that if one can change the training data distribution such that learning shares similar properties to that of training with SAM, then the new distribution can drive (S)GD and even SAM toward finding more generalizable solutions.

To address the above question, we first theoretically analyze the dynamics of training a two-layer convolutional neural network (CNN) with SAM and compare it with that of GD. We rigorously prove that SAM learns different features in a more *uniform speed* compared to GD, particularly *early* in training. In other words, we show that *SAM is less susceptible to simplicity bias* than GD. Simplicity bias of SGD makes the model learn simple solutions with minimum norm [25] and has long been conjectured to be the reason for the superior generalization performance of overparameterized models by providing implicit regularization [7, 25, 28, 51, 52, 70]. Nevertheless, the minimum-norm solution found by GD can have a suboptimal performance [64].

Following our theoretical results, we formulate changing the distribution of a training dataset such that different features are learned at a more uniform speed. First, we prove that the model output for examples containing features that are learned early by GD is separable from the rest of examples in their class. Then, we propose changing the data distribution by (i) identifying a cluster of examples with similar model output early in training, (ii) upsampling the remaining examples once to speed up their learning, and (iii) restarting training on the modified training distribution. Our method, UpSample Early For Uniform Learning (USEFUL), effectively alleviates the simplicity bias and consequently improves the generalization performance. Intuitively, learning features in a more uniform speed prevents the model to overfit underrepresented but useful features that otherwise are learned in late training stages. When the model overfits an example, it cannot learn its features in a generalizable manner. This harms the generalization performance on the original data distribution.

We show the effectiveness of USEFUL in alleviating the simplicity bias and improving the generalization via extensive experiments. First, we show that despite being relatively lightweight, USEFUL effectively improves the generalization performance of SGD and SAM. Additionally, we show that USEFUL can be easily applied with various optimizers and data augmentation methods to improve in-distribution generalization performance even further. For example, applying USEFUL with SAM and TrivialAugment (TA) [50] achieves, to the best of our knowledge, *state-of-the-art* accuracy for image classification for training ResNet18 on CIFAR10, STL10, CINIC10, Tiny-ImageNet; ResNet34 on CIFAR100; and VGG19 and DenseNet121 on CIFAR10. We also empirically confirm the benefits of USEFUL to out-of-distribution performance, but we emphasize that this is not the focus of our work.

## 2 Related Works

**Sharpness-aware-minimization (SAM).** Motivated by the generalization advantages of flat local minima, sharpness-aware minimization (SAM) was concurrently proposed in [22, 81] to minimize the training loss at the worst perturbed direction from the current parameters. SAM has been shown to obtain state-of-the-art on a variety of tasks [22]. Additionally, SAM has been shown to be beneficial in other settings, including label noise [22, 81], and domain generalization [9, 72].

There have been recent efforts to understand the generalization benefits of SAM. The most popular explanation is based on the Hessian spectra, empirically [22, 36] and theoretically [6, 73]. Other works showed that SAM finds a sparser solution in diagonal linear networks [3], and exhibits benign overfitting under much weaker signal strength compared to (S)GD [12]. More recently, SAM is shown to also benefit out-of-distribution (OOD). In particular, [65] suggested that SAM promotes diverse feature learning by empirically studying a simplified version of SAM which only perturbs the last layer. They showed that SAM upscales the last layer's weights to induce feature diversity, which benefits OOD. In contrast, we rigorously analyze a 2-layer non-linear CNN and prove that SAM learns (the same set of) features at a more uniform speed, which benefits the in-distribution (ID) settings. Our results reveal an orthogonal effect of SAM that benefits the ID generalization by reducing the simplicity bias, and provides a complementary view to prior works explaining superior ID generalization performance of SAM. We then propose a method to learn features more evenly by changing the data distribution.

**Simplicity bias (SB).** (S)GD has an inductive bias towards learning simpler solutions with minimum norm [25]. It is empirically observed [34] and theoretically proved [28] that SGD learns linear functions in the early training phase and more complex functions later in training. SB of SGD has been long conjectured to be the reason for the superior in-distribution generalization performance of overparameterized models, by providing capacity control or implicit regularization [26, 52, 55, 63]. On the other hand, in the OOD setting, simplicity bias is known to contribute to shortcut learning by causing models to exclusively rely on the simplest spurious feature and remain invariant to the complex but more predictive features [63, 67, 75]. Prior works on mitigating simplicity bias have been shown effective in the OOD settings [67, 68]. In contrast, our work shows, for the first time, that reducing the simplicity bias also benefits the ID settings. By studying the mechanism of feature learning in a two-layer nonlinear CNN, we prove that SAM is less susceptible to simplicity bias than GD, in particular early in training, which contributes to its superior performance. Then, we show that training data distribution can be modified to reduce the SB and improve the in-distribution generalization. In Appendix D.7, we empirically confirm that existing simplicity bias mitigation methods also improve the in-distribution performance, but to a smaller extent than ours.

**Distinction from Existing Settings.** Our work is distinct from the following literature:

(1) *Distribution Shift.* Unlike distribution shift and shortcut learning [18, 39, 57, 61], we *do not* assume existence of domain-dependent (non-generalizable) features or strong spurious correlations in the training data, or shift between training and test distribution. We focus on *in-distribution* generalization, where training and test distributions are the same and all the features in the training data are relevant for generalization. In Appendix D.5 we empirically show the benefits of our method to distribution shift, but we emphasize that this is not the focus of our study and we leave this direction to future work.

(2) *Long-tail distribution.* Long-tailed data is studied as a special case of distribution shift in which (sub)classes are highly imbalanced in training but are (more) balanced in test data [15, 71]. Long-tail methods resample the data at the class or subclass level to match the training and test distribution. In contrast, in our settings, training and test data follow the same distribution. Nevertheless, our method can be applied to improve the performance of long-tail datasets, as we confirm in Appendix D.5.

(3) *Improving Convergence.* A body of work speeds up convergence of (S)GD to find the *same solution* faster. Such methods iteratively sample or reweight examples based on loss or gradient norm during training [21, 33, 35, 80]. In contrast, our work does not intend to speed up training to find the same solution faster, but intends to find a *more generalizable solution* on the original data distribution.

(4) *Data Filtering Methods.* Filtering methods identify and discard or downweight noisy labeled [45], domain mismatched [23], redundant [1, 44, 59], or adversarial examples crafted by data poisoning attacks [66]. In contrast, we assume a *clean* training data and no mismatch between training and test distribution. Our work can be applied to a filtered training data to further improve the performance.

## 3  Theoretical Analysis: SAM Learns Different Features More Evenly

In this section, we analyze and compare feature learning mechanism of SAM. First, we introduce our theoretical settings including data distribution and neural network model in Sec. 3.1. We then revisit the update rules of GD and SAM in Sec. 3.2 before presenting our theoretical results in Sec. 3.3.

### 3.1  Theoretical Settings

**Notation.** We use lowercase letters, lowercase boldface letters, and uppercase boldface letters to denote scalars ($a$), vectors ($\boldsymbol{v}$), and matrices ($\boldsymbol{W}$). For a vector $\boldsymbol{v}$, we use $\|\boldsymbol{v}\|_2$ to denote its Euclidean norm. Given two sequence $\{x_n\}$ and $\{y_n\}$, we denote $x_n = O(y_n)$ if $|x_n| \leq C_1 |y_n|$ for some absolute positive constant $C_1$, $x_n = \Omega(y_n)$ if $|x_n| \geq C_2 |y_n|$ for some absolute positive constant $C_2$, and $x_n = \Theta(y_n)$ if $C_3 |y_n| \leq |x_n| \leq C_4 |y_n|$ for some absolute constant $C_3, C_4 > 0$. Besides, we use $\tilde{O}(\cdot), \tilde{\Omega}(\cdot)$, and $\tilde{\Theta}(\cdot)$ to hide logarithmic factors in these notations. Furthermore, we denote $x_n = \text{poly}(y_n)$ if $x_n = O(y_n^D)$ for some positive constant D, and $x_n = \text{polylog}(y_n)$ if $x_n = \text{poly}(\log(y_n))$.

**Data distribution.** We use a popular data distribution used in recent works on feature learning [2, 8, 11, 12, 18, 32, 40] to represent data as a combination of two features and noise patches. Additionally, we introduce a probability $\alpha$ to control the frequency of fast-learnable features in the data distribution.

**Definition 3.1** (Data distribution). A data point $(\boldsymbol{x}, y) \in (\mathbb{R}^d)^P \times \{\pm 1\}$ is generated from the distribution $\mathcal{D}(\beta_e, \beta_d, \alpha)$ as follows. We uniformly generate the label $y \in \{\pm 1\}$. We generate $\boldsymbol{x}$ as a collection of $P$ patches: $\boldsymbol{x} = (\boldsymbol{x}^{(1)}, \boldsymbol{x}^{(2)}, \dots, \boldsymbol{x}^{(P)}) \in (\mathbb{R}^d)^P$, where

- **Slow-learnable Feature.** One and only one patch is given by $\beta_d \cdot y \cdot \boldsymbol{v}_d$ with $\|\boldsymbol{v}_d\|_2 = 1$, $\langle \boldsymbol{v}_e, \boldsymbol{v}_d \rangle = 0$, and $0 \le \beta_d < \beta_e \in \mathbb{R}$.

- **Fast-learnable feature.** One and only one patch is given by $\beta_e \cdot y \cdot \boldsymbol{v}_e$ with $\|\boldsymbol{v}_e\|_2 = 1$ with a probability $\alpha \le 1$. With a probability of $1 - \alpha$, this patch is masked, i.e. $\boldsymbol{0}$.

- **Random noise.** The rest of $P - 2$ patches are Gaussian noise $\boldsymbol{\xi}$ that are independently drawn from $N(0, (\sigma_p^2/d) \cdot \mathbf{I}_d)$ with $\sigma_p$ as an absolute constant.

For simplicity, we assume $P = 3$, and the noisy patch together with two features form an orthogonal set. Coefficients $\beta_e$ and $\beta_d$ characterize the feature strength in our data model. A larger coefficient means that the corresponding feature is learned faster.

**Two-layer nonlinear CNN.** To model modern state-of-the-art architectures, we analyze a two-layer nonlinear CNN which is also used in [8, 11, 18, 32, 40]. Unlike linear models, CNN can handle a data distribution that does not require a fixed position of patches as defined above. Formally,

$$f(\boldsymbol{x}; \boldsymbol{W}) = \sum_{j \in [J]} \sum_{p=1}^{P} \sigma(\langle \boldsymbol{w}_j, \boldsymbol{x}^{(p)} \rangle), \tag{1}$$

where $\boldsymbol{w}_j \in \mathbb{R}^d$ is the weight vector of the $j$-th filter, $J$ is the number of filters (neurons) of the network, and $\sigma(z) = z^3$ is the activation function, i.e., the main source of non-linearity. $\boldsymbol{W} = [\boldsymbol{w}_1, \dots, \boldsymbol{w}_J] \in \mathbb{R}^{d \times J}$ is the weight matrix of the CNN. Following [8, 18, 32], we assume a mild over-parameterization with $J = \text{polylog}(d)$. We initialize $\boldsymbol{W}^{(0)} \sim \mathcal{N}(0, \sigma_0^2)$, where $\sigma_0^2 = \text{polylog}(d)/d$.

## 3.2 Empirical Risk Minimization: GD vs SAM

Consider a $N$-sample training dataset $D = \{(\boldsymbol{x}_i, y_i)\}_{i=1}^N$ in which each data point is generated from the data distribution in Definition 3.1. The empirical loss function of a model $f(\boldsymbol{x}; \boldsymbol{W})$ reads

$$\mathcal{L}(\boldsymbol{W}) = \frac{1}{N} \sum_{i=1}^{N} l(y_i f(\boldsymbol{x}_i; \boldsymbol{W})), \tag{2}$$

where $l$ is the logistic loss defined as $l(z) = \log(1 + \exp(-z))$. The solution $\boldsymbol{W}^\star$ of the empirical risk minimization (ERM) minimizes the above loss, i.e., $\boldsymbol{W}^\star := \arg\min_{\boldsymbol{W}} \mathcal{L}(\boldsymbol{W})$.

**GD.** Typically, ERM is solved using gradient descent (GD). The update rule at iteration $t$ of GD with learning rate $\eta > 0$ reads

$$\boldsymbol{W}^{(t+1)} = \boldsymbol{W}^{(t)} - \eta \nabla \mathcal{L}(\boldsymbol{W}^{(t)}). \tag{3}$$

**SAM.** To find solutions with better generalization performance, Foret et al. [22] proposed the $N$-SAM algorithm that minimizes both loss and curvature. SAM's update rule at iteration $t$ reads

$$\boldsymbol{W}^{(t+1)} = \boldsymbol{W}^{(t)} - \eta \nabla \mathcal{L}(\boldsymbol{W}^{(t)} + \rho^{(t)} \nabla \mathcal{L}(\boldsymbol{W}^{(t)})), \tag{4}$$

where $\rho^{(t)} = \rho > 0$ is the inner step size that is usually normalized by gradient norm, i.e., $\rho^{(t)} = \rho / \|\nabla \mathcal{L}(\boldsymbol{W}^{(t)})\|_F$.

## 3.3 Comparing Learning Between fast-learnable & slow-learnable Features for GD & SAM

Next, we present our theoretical results on training dynamics of the two-layer nonlinear CNN using GD and SAM. We characterize the learning speed of features by studying the growth of the model outputs before the activation function, i.e., $\langle \boldsymbol{w}_j^{(t)}, \boldsymbol{v}_e \rangle$ and $\langle \boldsymbol{w}_j^{(t)}, \boldsymbol{v}_d \rangle$. We first prove that *early* in training, both GD and SAM *only* learn fast-learnable feature. Then, we show SAM learns slow-learnable and fast-learnable features at a more uniform speed.

**Theorem 3.2** (**GD Feature Learning**). *Consider training a two-layer nonlinear CNN model initialized with $\boldsymbol{W}^{(0)} \sim \mathcal{N}(0, \sigma_0^2)$ on the training dataset $D = \{(\boldsymbol{x}_i, y_i)\}_{i=1}^N$ with distribution $\mathcal{D}(\beta_e, \beta_d, \alpha)$ with $\alpha^{1/3}\beta_e > \beta_d$. For a small-enough learning rate $\eta$, after training for $T_{GD}$ iterations, w.h.p., the model: (1) learns the fast-learnable feature $\boldsymbol{v}_e$: $\max_{j \in [J]} \langle \boldsymbol{w}_j^{(T_{GD})}, \boldsymbol{v}_e \rangle \geq \tilde{\Omega}(1/\beta_e)$; (2) does not learn the slow-learnable feature $\boldsymbol{v}_d$: $\max_{j \in [J]} \langle \boldsymbol{w}_j^{(T_{GD})}, \boldsymbol{v}_d \rangle = \tilde{O}(\sigma_0)$.*

**Theorem 3.3** (**SAM Feature Learning**). *Consider training a two-layer nonlinear CNN model initialized with $\boldsymbol{W}^{(0)} \sim \mathcal{N}(0, \sigma_0^2)$ on the training dataset $D = \{(\boldsymbol{x}_i, y_i)\}_{i=1}^N$ with distribution $\mathcal{D}(\beta_e, \beta_d, \alpha)$ with $\alpha^{1/3}\beta_e > \beta_d$. For small-enough learning rate $\eta$ and perturbation radius $\rho$, after training for $T_{SAM} > T_{GD}$ iterations, w.h.p., the model: (1) learns the fast-learnable feature $\boldsymbol{v}_e$ : $\max_{j \in [J]} \langle \boldsymbol{w}_j^{(T_{SAM})}, \boldsymbol{v}_e \rangle \geq \tilde{\Omega}(1/\beta_e)$; (2) does not learn the slow-learnable feature $\boldsymbol{v}_d$ : $\max_{j \in [J]} \langle \boldsymbol{w}_j^{(T_{SAM})}, \boldsymbol{v}_d \rangle = \tilde{O}(\sigma_0)$.*

The detailed proof of Theorems 3.2 and 3.3 are deferred to Appendices A.1 and A.2.

**Discussion.** Note that a larger value of $\langle \boldsymbol{w}_j^{(t)}, \boldsymbol{v} \rangle$ for $\boldsymbol{v} \in \{\boldsymbol{v}_e, \boldsymbol{v}_d\}$ indicates better learning of the feature vector $\boldsymbol{v}$ by neuron $\boldsymbol{w}_j$ at iteration $t$. From the above two theorems, the growth rate of the fast-learnable feature is significantly faster than that of the slow-learnable feature. As a small portion $(1 - \alpha)$ of the dataset does not have the fast-learnable feature, the model needs to learn the slow-learnable feature to improve the performance.

Next, we show that SAM learns fast-learnable and slow-learnable features more evenly. We denote by $G_e^{(t)} = \max_{j \in [J]} \langle \boldsymbol{w}_j^{(t)}, \boldsymbol{v}_e \rangle$ and $G_d^{(t)} = \max_{j \in [J]} \langle \boldsymbol{w}_j^{(t)}, \boldsymbol{v}_d \rangle$ the alignment of model weights with fast-learnable and slow-learnable features, when training with GD. Similarly, we denote by $S_e^{(t)}$ and $S_d^{(t)}$ the alignment of model weights with fast-learnable and slow-learnable , when training with SAM.

**Theorem 3.4** (**SAM learns features more evenly than GD**). *Consider the same model and training dataset as Theorems 3.2 and 3.3. Assume that the learning rate $\eta$ and the perturbation radius $\rho$ are sufficiently small. Starting from the same initialization, the growth of fast-learnable and slow-learnable features in SAM is more balanced than that in SGD, i.e., for every iteration $t \in [1, T_0]$:*

$$S_e^{(t)} - S_d^{(t)} < G_e^{(t)} - G_d^{(t)}. \tag{5}$$

We prove Theorem 3.4 by induction in Appendix A.2 and back it by toy experiments in Section 5.1.

**Discussion.** Intuitively, our proof is based on the fact that the difference between the growth of fast-learnable and slow-learnable features in SAM is smaller than that of GD. Thus, starting from the same initialization, the slow-learnable feature contributes relatively more to the model prediction in SAM than it does in SGD. Thus, the slow-learnable feature benefits SAM, by reducing its overreliance on the fast-learnable features. We note that as neural networks are nonlinear, a small change in the output can actually result in a big change in the model and its performance. Even in the extreme setting when two features have identical strength and the fast-learnable feature exists in all examples, i.e., $\beta_e = \beta_d = \alpha = 1$, the gap in Eq. 5 is significant as we confirm in Figure 8 in Appendix D.

**Remark.** The network often overfits slow-learnable features that are learned late during the training and do not learn them in a generalizable manner. This harms the generalization performance on the test set sampled from the *original* data distribution.

Theorems 3.2 and 3.3 show that we can make the model learn more from the slow-learnable feature by increasing the value of $\beta_d$. Based on this intuition, we have the following theorem.

**Theorem 3.5** (**One-shot upsampling**). *Under the assumptions of Theorems 3.2, 3.3, for a sufficiently small noise, from any iteration $t$ during early training, we have the following results:*

1. *The slow-learnable feature has a larger contribution to the normalized gradient of the 1-step SAM update, compared to that of GD.*

2. *Amplifying the strength of the slow-learnable feature increases its contribution to the normalized gradients of GD and SAM.*

3. *There exists an upsampling factor $k$ s.t. the normalized gradient of the 1-step GD update on $\mathcal{D}(\beta_e, k\beta_d, \alpha)$ recovers the normalized gradient of the 1-step SAM update on $\mathcal{D}(\beta_e, \beta_d, \alpha)$.*

**Algorithm 1** UpSample Early For Uniform Learning (USEFUL)

---

**Input:** Original dataset $D$, Model $f(\cdot, \boldsymbol{W}^{(0)})$, Separating epoch $t$, Total epochs $T$.
Train the model $f(\cdot, \boldsymbol{W}^{(0)})$ on $D$ for $t$ epochs.
**for** every class $c \in D$ **do**
$\quad \{C_1, C_2\} \leftarrow k\text{-means}(f(\boldsymbol{x}_j; \boldsymbol{W}^{(t)}))$
$\quad D = D \cup C_2$, where $C_2$ is the cluster with higher average loss
**end for**
Train $f(\cdot, \boldsymbol{W}^{(0)})$ on $D$ for $T$ epochs
**Output:** Model $f(\cdot, \boldsymbol{W}^{(T)})$

---

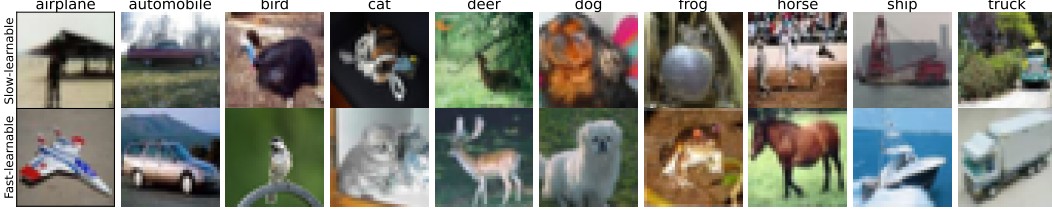

Figure 1: Examples of slow-learnable (top) and fast-learnable (bottom) in CIFAR-10 found by our method. Examples in the top row (slow-learnable) are harder to identify visually and look more ambiguous (part of the object is in the image or the object is smaller and the area associated with the background is larger). In contrast, examples in the bottom row (fast-learnable) are not ambiguous and are clear representatives of their corresponding class, hence are very easy to visually classify (the entire object is in the image and the area associated with the background is small).

**Discussion.** Proof of Theorem 3.5 is given in Appendix A.4. We see that we can learn features at a more uniform speed by training on a new dataset $\mathcal{D}(\beta_e, \beta_d', \alpha)$ with a larger strength $\beta_d' > \beta_d$. But, the value of coefficient $\beta_d'$ varies by the model weights and gradient at each iteration $t$.

**Remark.** Intuitively, Theorem 3.5 implies that by descending over a flatter trajectory, SAM learns slow-learnable features relatively earlier in training, compared to GD. While the largest difference between feature learning of SAM and (S)GD is attributed to early training dynamics (due to the simplicity bias of (S)GD and its largest contribution early in training), SAM learns features at a more uniform speed during the *entire* training. This effect is, however, difficult to theoretically characterize exactly. Therefore, learning features at a more uniform speed via SAM help yield flatter minima with better generalization performance. We note that while SAM learns fast-learnable and slow-learnable features at a more uniform speed, it still suffers from simplicity bias and learns fast-learnable features earlier (although less so than GD) as evidenced in our Theorem 3.3.

## 4 Method: UpSample Early For Uniform Learning (USEFUL)

Motivated by our theoretical results, we aim to speed up learning the slow-learnable features in the training data. This drive the network to learn fast-learnable and slow-learnable features at a more uniformly speed, and ultimately improves the in-distribution generalization performance.

**Step 1: Identifying examples with fast-learnable features.** As shown in Theorems 3.2 and 3.3, fast-learnable features are learned early in training, and the *model output for examples containing fast-learnable features are highly separable* from the rest of examples in their class, early in training. This is illustrated for one class of a toy example and CIFAR-10 in Fig. 2. Motivated by our theory, we seek to find a cluster of examples with similar model outputs early in training. To do so, we apply $k$-means clustering to the last-layer activation vectors of examples in every class, to separate examples with fast-learnable features from the rest of examples. Formally, for examples in every class with $y_j = c$, we find:

$$\arg\min_C \sum_{i \in \{1,2\}} \sum_{y_j = c, j \in C_i} \|f(\boldsymbol{x}_j; \boldsymbol{W}^{(t)}) - \boldsymbol{\mu}_i\|^2, \tag{6}$$

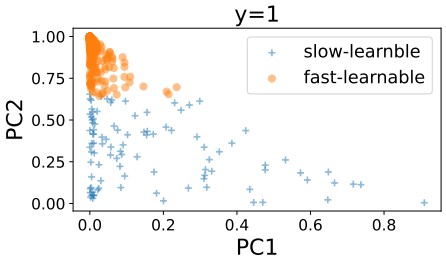
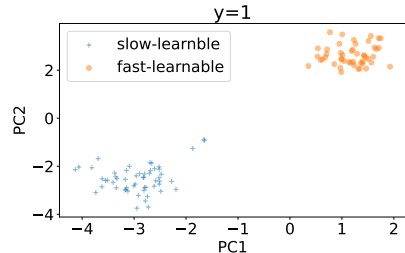

Figure 2: TSNE visualization of output vectors. (left) ResNet18/CIFAR-10 at epoch 8. (right) CNN/toy data generated based on Definition 3.1 with $\beta_d = 0.2, \beta_e = 1, \alpha = 0.9$, iteration 200.

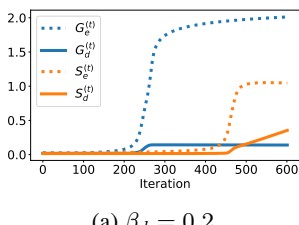
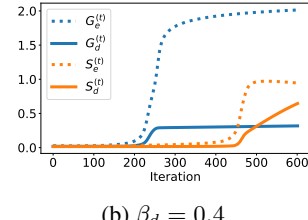
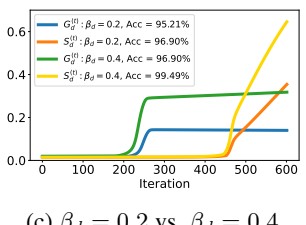

(a) $\beta_d = 0.2$        (b) $\beta_d = 0.4$        (c) $\beta_d = 0.2$ vs. $\beta_d = 0.4$

Figure 3: **GD (blue) vs. SAM (orange) on toy datasets.** Data is generated based on Definition 3.1 with different $\beta_d$ and fixed $\beta_e = 1, \alpha = 0.9$. $\cdot\cdot$ and $--$ lines denote the alignment (i.e., inner product) of fast-learnable ($\boldsymbol{v}_e$) and slow-learnable ($\boldsymbol{v}_d$) features with the model weight ($\boldsymbol{w}_j^{(t)}$). (a), (b) GD and SAM first learn the fast-learnable feature. Notably, GD learns the fast-learnable feature very early. (c) Test accuracy of GD & SAM improves by increasing the strength of the slow-learnable feature.

where $\boldsymbol{\mu}_i$ is the center of cluster $S_i$. The cluster with lower average loss will contain the majority of examples containing fast-learnable features, whereas the remaining examples contain slow-learnable features in the training data. Examples of images in fast-learnable and slow-learnable clusters of CIFAR-10 found by USEFUL are illustrated in Fig. 1.

**The choice of clustering.** Our choice of clustering is motivated by our Theorems 3.2 and 3.3 which show that examples with fast-learnable features are separable based on model output from the rest of examples in their class. While examples with fast-learnable features are expected to have a lower loss, loss of examples may oscillate during the training and makes it difficult to find an accurate cut-off for separating the examples. Besides, as fast-learnable features may not be *fully* learned early in training, examples containing fast-learnable features may not necessarily have the right prediction, thus misclassification cannot separate examples accurately. In contrast, clustering does not require hyperparameter tuning and performs well for separating examples, as we confirm in our ablation studies.

**Step 2: One-shot upsampling of slow-learnable features.** Next, we upsample examples that are not in the cluster of points containing fast-learnable features. This speeds up learning slow-learnable features and encourages the model to learn different features at a more uniform speed. Thus, it improves the in-distribution performance based on Theorem 3.5. As discussed earlier, the number of times we upsample these examples should change based on the model weight at each iteration. Hence, a multi-stage clustering and sampling can yield the best results. Nevertheless, we empirically confirm that a 1-shot algorithm that finds fast-learnable examples at an *early* training iteration and upsample the remaining examples by a factor of $k = 2$ effectively improves the performance. Notably, in contrast to dynamic sampling or reweighting, USEFUL upsamples examples only once and restart training on the modified but fix distribution.

**When to separate the examples.** It is crucial to separate examples *early* in training, to accurately identify examples that contribute the most to simplicity bias. We empirically verify the intuition that the optimal epoch $t$ to separate examples is when the change in training error starts to shrink as visualized in Figure 15a. More details can be found in Appendices C.2 and D.8.

The pseudocode of USEFUL is illustrated in Alg. 1 and the workflow is shown in Appendix Fig. 7.

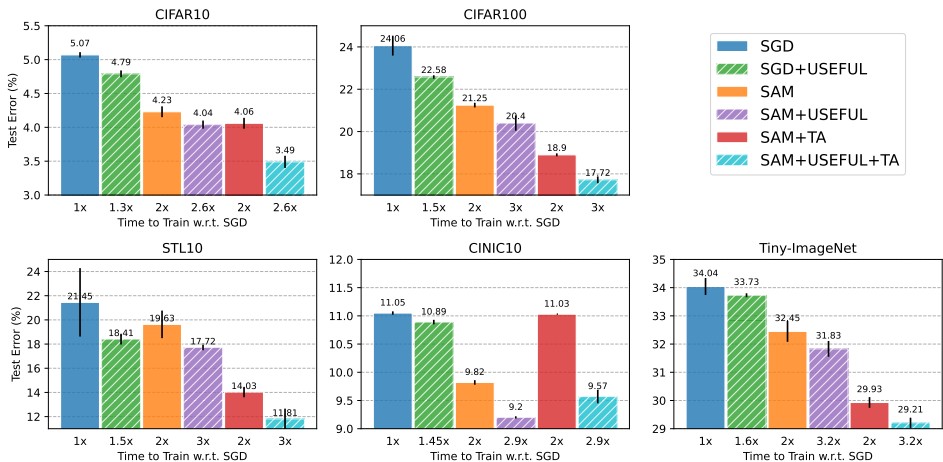

Figure 4: **Test classification error of ResNet18 on CIFAR10, STL10, TinyImageNet and ResNet34 on CIFAR100.** The numbers below bars indicate the approximate training cost and the tick on top shows the std over three runs. USEFUL enhances the performance of SGD and SAM on all 5 datasets. TrivialAugment (TA) further boosts SAM's performance (except for CINIC10). Remarkably, USEFUL consistently boosts the performance across all scenarios and achieves (to our knowledge) SOTA performance for ResNet18 and ResNet34 on the selected datasets when combined with SAM and TA.

## 5 Experiments

**Outline.** In Sec. 5.1, we empirically validate our theoretical results on toy datasets. We then evaluate the performance of USEFUL on several real-world datasets in Sec. 5.2 and different model architectures in Sec. 5.3. In addition, Sec. 5.3 highlights the advantages of USEFUL over random upsampling. Furthermore, we show that USEFUL shares several properties with SAM in Sec. 5.4. Additional experimental results are deferred to Appendix D where we show that USEFUL also boosts the performance of other SAM variants, and present promising results for USEFUL applied to the OOD setting (spurious correlation, long-tail distribution), transfer learning, and label noise settings. We further conduct ablation studies on the effect of our data selection strategy for upsampling, training batch size, learning rate, upsampling factor, and separating epoch in Appendix D.8.

**Settings.** We used common datasets for image classification including CIFAR10, CIFAR100 [41], STL10 [13], CINIC10 [16], and Tiny-ImageNet [43]. Both CINIC10 and Tiny ImageNet are large-scale datasets containing images from the ImageNet dataset [17]. We trained ResNet18 on all datasets except for CIFAR100 on which we trained ResNet34. We closely followed the setting from [3] in which our models are trained for 200 epochs with a batch size of 128. We used SGD with the momentum parameter of 0.9 and set weight decay to 0.0005. We also fixed $\rho = 0.1$ for SAM in all experiments unless explicitly stated. We used a linear learning rate schedule starting at 0.1 and decay by a factor of 10 once at epoch 100 and again at epoch 150. More details are given in Appendix C.

### 5.1 Toy Datasets

**Datasets.** Following [18], our toy dataset consists of training and test sets, each containing 10K examples generated from the data distribution defined in 3.1 with dimension $d = 50$ and $P = 3$. We set $\beta_e = 1, \beta_d = 0.2, \alpha = 0.9$, and $\sigma_p/\sqrt{d} = 0.125$. We also consider a scenario with larger $\beta_d = 0.4$. We shuffle the order of patches randomly to confirm that our theory holds with arbitrary order of patches.

**Training.** We used the two-layer nonlinear CNN in Section 3.1 with $J = 40$ filters. For GD, we set the learning rate to $\eta = 0.1$ and did not use momentum. For SAM, we used the same base GD optimizer and chose a smaller value of the inner step, $\rho = 0.02$, than other experiments to satisfy the constraint in Theorem 3.4. We trained the model for 600 iterations till convergence for GD and SAM.

**Results.** Figure 3a illustrates that both GD (blue) and SAM (orange) first learn the fast-learnable feature. In particular, the blue dotted line ($G_e$) accelerates quickly at around epoch 250 while the orange dotted line ($S_e$) increases drastically much later at around epoch 450. That is: **_(1) GD learns the fast-_**

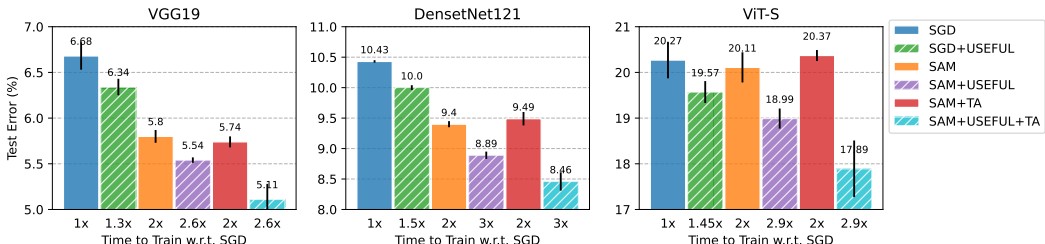

Figure 5: **Test classification errors of different architectures on CIFAR10.** USEFUL improves the performance of SGD and SAM when training different architectures. TrivialAugment (TA) further boosts SAM's capabilities. The results for 3-layer MLP can be found in Figure 9.

***learnable feature very early in training.*** This is well-aligned with our Theorems 3.2 and 3.3 and their discussion. Furthermore, the gap between contribution of fast-learnable and slow-learnable features towards the model output in SAM ($S_e^{(t)} - S_d^{(t)}$) is much smaller than that of GD ($G_e^{(t)} - G_d^{(t)}$). That is: ***(2) fast-learnable and slow-learnable features are learned more evenly in SAM.*** This validates our Theorem 3.4. From around epoch 500 onwards, the contribution of the slow-learnable feature in SAM surpasses the level of that in GD while the contribution of the fast-learnable feature in SAM is still lower than the counterpart in GD. When increasing the slow-learnable feature strength $\beta_d$ from 0.2 to 0.4 in Figure 3b, the same conclusion for the growth speed of fast-learnable and slow-learnable features holds. Notably, there is a clear increase in the classification accuracy of the model trained with either GD or SAM by increasing $\beta_d$, as can be seen in Figure 3c. That is: ***(3) amplifying the strength of the slow-learnable feature improves the generalization performance.*** Effectively, this enables the model successfully predict examples in which the fast-learnable feature is missing.

## 5.2 USEFUL is Effective across Datasets

Figure 4 illustrates the performance of models trained with SGD and SAM on original vs modified data distribution by USEFUL. We see that USEFUL effectively reduces the test classification error of both SGD and SAM. Interestingly, USEFUL further improves SAM's generalization performance by reducing its simplicity bias. Notably, on the STL10 dataset, USEFUL boosts the performance of SGD to surpass that of SAM. The percentages of examples found for upsampling by USEFUL for CIFAR10, CIFAR100, STL10, CINIC10, and Tiny-ImageNet are roughly 30%, 50%, 50%, 45%, and 60%, respectively. Thus, training SGD on the modified data distribution only incurs a cost of 1.3x, 1.5x, 1.5x, 1.45x, and 1.6x compared to 2x of SAM.

**USEFUL+TA is particularly effective.** Stacking strong augmentation methods e.g. TrivialAugment [50] further improves the performance, achieving state-of-the-art for ResNet on all datasets. When strong augmentation is combined with USEFUL, it makes more variations of the (upsampled) slow-learnable features and enhances their learning. Hence, it further boost the performance.

## 5.3 USEFUL is Effective across Architectures & Settings

**Model architectures: CNN, ViT, MLP.** Next, we confirm the versatility of our method, by applying it to different model architectures including 3-layer MLP, CNNs (ResNet18, VGG19, DenseNet121), and Transformers (ViT-S). Figure 5 shows that USEFUL is effective across different model architectures. Remarkably, when applying to non-CNN architectures, it reduces the test error of SGD to a lower level than that of SAM alone. Detailed results for 3-layer MLP is given in Appendix D.2.

**Settings: batch-size, learning rate, and SAM variants.** In Appendix D, we confirm the effectiveness of USEFUL for different batch sizes of 128, 256, 512, and different initial learning rates of 0.1, 0.2, 0.4. In Appendix D.4, we confirm that USEFUL applied to ASAM [42]—a SAM variant which uses a scale-invariant sharpness measure—further reduces the test error.

**USEFUL vs Random Upsampling.** Fig. 6 shows that USEFUL considerably outperforms SGD and SAM on randomly upsampled CIFAR10 & CIFAR100. This confirms that the main benefit of USEFUL is due to the modified distribution and not longer training time. In Appendix D.8, we also confirm that upsampling outperforms upweighting for SAM & SGD.

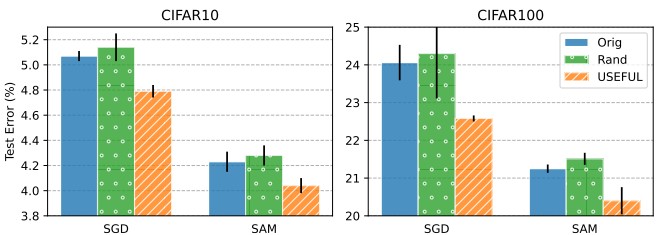

Figure 6: USEFUL vs. Random Upsampling, when training ResNet18 on CIFAR10 and CIFAR100.

### 5.4    USEFUL's Solution has Similar Properties to SAM

**SAM & USEFUL Find Sparser Solutions than SGD.** [3] showed that SAM's solution has a better sparsity-inducing property indicated by the L1 norm than the standard ERM. Fig. 10 shows the L1 norm of ResNet18 trained on CIFAR10 and ResNet34 trained on CIFAR100 at the end of training. We see that USEFUL drives both SGD and SAM to find solutions with smaller L1 norms.

**SAM & USEFUL Find Less Sharp Solutions than SGD.** While our goal is not to directly find a flatter minimum or the same solution as SAM, we showed that USEFUL finds flatter minima. Following [3], we used the maximum Hessian eigenvalue ($\lambda_{max}$) and the bulk of the spectrum ($\lambda_{max}/\lambda_5$) [30], which are commonly used metrics for sharpness [10, 31, 37, 74]. Table 1 illustrates that SGD+USEFUL on CIFAR10 reduces sharpness metrics significantly compared to SGD, proving that USEFUL successfully reduces the sharpness of the solution. We note that to capture the sharpness/flatness, multiple different criteria have been proposed (largest Hessian eigenvalue and bulk of Hessian), and one criterion is not enough to accurately capture the sharpness. While the solution of SGD+USEFUL has a higher largest Hessian eigenvalue than SAM, it achieves the smallest bulk.

**SAM & USEFUL Reduce Forgetting Scores.** Forgetting scores [69] count the number of times an example is misclassified after being correctly classified during training and is an indicator of the learning speed and difficulty of examples. We show in Appendix D.3 that both SAM and USEFUL successfully reduce the forgetting scores, thus learn slow-learnable features faster than SGD. This aligns with our Theorem 3.4 and results on the toy datasets. By upsampling slow-learnable examples in the dataset, they contribute more to learning and hence SGD+USEFUL learns them faster than SGD.

**USEFUL also Benefits Distribution Shift.** While our main contribution is providing a novel and effective method to improve the in-distribution generalization performance, we conduct experiments confirming the benefits of our method to distribution shift. We discuss this experiment and its results in Appendix D.5. On Waterbirds dataset [61] with strong spurious correlation (95%), both SAM and USEFUL successfully improve the performance on the balanced test set by 6.21% and 5.8%, respectively. We also show the applicability of USEFUL to fine-tuning a ResNet50 pre-trained on ImageNet.

## 6    Conclusion

In this paper, we made the first attempt to improve the in-distribution generalization performance of machine learning methods by modifying the distribution of training data. We first analyzed learning dynamics of sharpness-aware minimization (SAM), and attributed its superior performance over GD to mitigating the simplicity bias, and learning features at a more speed. Inspired by SAM, we upsampled the examples that contain slow-learnable features to alleviate the simplicity bias. This allows learning features more uniformly, thus improving the performance. Our method boosts the performance of image classifiers trained with SGD or SAM and easily stacks with data augmentation.

## Acknowledgments

This research was partially supported by the National Science Foundation CAREER Award 2146492, National Science Foundation 2421782 and Simons Foundation, Cisco Systems, Optum AI, and a UCLA Hellman Fellowship.

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

# A   Formal Proofs

## A.1   Proof of Theorem 3.2

**Notation.** In this paper, we use lowercase letters, lowercase boldface letters, and uppercase boldface letters to respectively denote scalars ($a$), vectors ($\boldsymbol{v}$), and matrices ($\boldsymbol{W}$). For a vector $\boldsymbol{v}$, we use $\|\boldsymbol{v}\|_2$ to denote its Euclidean norm. Given two sequence $\{x_n\}$ and $\{y_n\}$, we denote $x_n = O(y_n)$ if $|x_n| \leq C_1|y_n|$ for some absolute positive constant $C_1$, $x_n = \Omega(y_n)$ if $|x_n| \geq C_2|y_n|$ for some absolute positive constant $C_2$, and $x_n = \Theta(y_n)$ if $C_3|y_n| \leq |x_n| \leq C_4|y_n|$ for some absolute constant $C_3, C_4 > 0$. In addition, we use $\tilde{O}(\cdot), \tilde{\Omega}(\cdot)$, and $\tilde{\Theta}(\cdot)$ to hide logarithmic factors in these notations. Furthermore, we denote $x_n = \text{poly}(y_n)$ if $x_n = O(y_n^D)$ for some positive constant D, and $x_n = \text{polylog}(y_n)$ if $x_n = \text{poly}(\log(y_n))$.

First, we have the following assumption for the model weight initialization.

**Assumption A.1** (Weight initialization). Assume that we initialize $\boldsymbol{W}^{(0)} \sim \mathcal{N}(0, \sigma_0^2)$ such that for all $j \in [J]$, $\langle \boldsymbol{w}_j^{(0)}, \boldsymbol{v}_e \rangle, \langle \boldsymbol{w}_j^{(0)}, \boldsymbol{v}_d \rangle \geq \rho > 0$.

The above assumption is reasonable because we later show that both sequences $\langle \boldsymbol{w}_j^{(t)}, \boldsymbol{v}_e \rangle$ and $\langle \boldsymbol{w}_j^{(t)}, \boldsymbol{v}_d \rangle$ are non-decreasing. So, we can obtain the above initialization by training the model for several iterations. For simplicity of the notation, we assume that $\alpha N$ is an integer and the first $\alpha N$ data examples have the fast-learnable feature while the rest do not. Before going into the analysis, we denote the derivative of a data example $i$ at iteration $t$ to be

$$l_i^{(t)} = \frac{\exp(-y_i f(\boldsymbol{x}_i; \boldsymbol{W}^{(t)}))}{1 + \exp(-y_i f(\boldsymbol{x}_i; \boldsymbol{W}^{(t)}))} = \text{sigmoid}(-y_i f(\boldsymbol{x}_i; \boldsymbol{W}^{(t)})). \tag{7}$$

**Lemma A.2** (Gradient). *Let the loss function $\mathcal{L}$ be as defined in Equation 2. For $t \geq 0$ and $j \in [J]$, the gradient of the loss $\mathcal{L}(\boldsymbol{W}^{(t)})$ with regard to neuron $\boldsymbol{w}_j^{(t)}$ is*

$$\nabla_{\boldsymbol{w}_j^{(t)}} \mathcal{L}(\boldsymbol{W}^{(t)}) = -\frac{3}{N} \sum_{i=1}^{\alpha N} l_i^{(t)} \left( \beta_d^3 \langle \boldsymbol{w}_j^{(t)}, \boldsymbol{v}_d \rangle^2 \boldsymbol{v}_d + \beta_e^3 \langle \boldsymbol{w}_j^{(t)}, \boldsymbol{v}_e \rangle^2 \boldsymbol{v}_e + y_i \langle \boldsymbol{w}_j^{(t)}, \boldsymbol{\xi}_i \rangle^2 \boldsymbol{\xi}_i \right) -$$

$$\frac{3}{N} \sum_{i=\alpha N+1}^{N} l_i^{(t)} \left( \beta_d^3 \langle \boldsymbol{w}_j^{(t)}, \boldsymbol{v}_d \rangle^2 \boldsymbol{v}_d + y_i \langle \boldsymbol{w}_j^{(t)}, \boldsymbol{\xi}_i \rangle^2 \boldsymbol{\xi}_i \right). \tag{8}$$

*Proof.* We have the following gradient

$$\nabla_{\boldsymbol{w}_j^{(t)}} \mathcal{L}(\boldsymbol{W}^{(t)}) = -\frac{1}{N} \sum_{i=1}^{N} \frac{\exp(-y_i f(\boldsymbol{x}_i; \boldsymbol{W}^{(t)}))}{1 + \exp(-y_i f(\boldsymbol{x}_i; \boldsymbol{W}^{(t)}))} \cdot y_i f'(\boldsymbol{x}_i; \boldsymbol{W}^{(t)})$$

$$= -\frac{3}{N} \sum_{i=1}^{N} l_i^{(t)} y_i \sum_{p=1}^{P} \langle \boldsymbol{w}_j^{(t)}, \boldsymbol{x}^{(p)} \rangle^2 \cdot \boldsymbol{x}^{(p)}$$

$$= -\frac{3}{N} \sum_{i=1}^{N} l_i^{(t)} \left( \beta_d^3 \langle \boldsymbol{w}_j^{(t)}, \boldsymbol{v}_d \rangle^2 \boldsymbol{v}_d + \beta_e^3 \langle \boldsymbol{w}_j^{(t)}, \boldsymbol{v}_e \rangle^2 \boldsymbol{v}_e + y_i \langle \boldsymbol{w}_j^{(t)}, \boldsymbol{\xi}_i \rangle^2 \boldsymbol{\xi}_i \right) -$$

$$\frac{3}{N} \sum_{i=\alpha N+1}^{N} l_i^{(t)} \left( \beta_d^3 \langle \boldsymbol{w}_j^{(t)}, \boldsymbol{v}_d \rangle^2 \boldsymbol{v}_d + y_i \langle \boldsymbol{w}_j^{(t)}, \boldsymbol{\xi}_i \rangle^2 \boldsymbol{\xi}_i \right)$$

$\square$

With the above formula of gradient, we have the following equations:

**Fast-learnable feature gradient.** The projection of the gradient on $\boldsymbol{v}_e$ is

$$\langle \nabla_{\boldsymbol{w}_j^{(t)}} \mathcal{L}(\boldsymbol{W}^{(t)}), \boldsymbol{v}_e \rangle = -\frac{3\beta_e^3}{N} \sum_{i=1}^{\alpha N} l_i^{(t)} \langle \boldsymbol{w}_j^{(t)}, \boldsymbol{v}_e \rangle^2 \tag{9}$$

**Slow-learnable feature gradient.** The projection of the gradient on $\boldsymbol{v}_d$ is

$$\langle \nabla_{\boldsymbol{w}_j^{(t)}} \mathcal{L}(\boldsymbol{W}^{(t)}), \boldsymbol{v}_d \rangle = -\frac{3\beta_d^3}{N} \sum_{i=1}^{N} l_i^{(t)} \langle \boldsymbol{w}_j^{(t)}, \boldsymbol{v}_d \rangle^2 \tag{10}$$

**Noise gradient.** The projection of the gradient on $\boldsymbol{\xi}_i$ is

$$\langle \nabla_{\boldsymbol{w}_j^{(t)}} \mathcal{L}(\boldsymbol{W}^{(t)}), \boldsymbol{\xi}_i \rangle = -\frac{3}{N} \left( l_i^{(t)} y_i \langle \boldsymbol{w}_j^{(t)}, \boldsymbol{\xi}_i \rangle^2 \|\boldsymbol{\xi}_i\|_2^2 + \sum_{k=1, k \neq i}^{N} l_k^{(t)} y_k \langle \boldsymbol{w}_j^{(t)}, \boldsymbol{\xi}_k \rangle^2 \langle \boldsymbol{\xi}_k, \boldsymbol{\xi}_i \rangle \right) \tag{11}$$

**Derivative of data example $i$.** For $1 \leq i \leq \alpha N$, $l_i^{(t)}$ can be rewritten as

$$l_i^{(t)} = \text{sigmoid} \left( \sum_{j=1}^{J} -\beta_d^3 \langle \boldsymbol{w}_j^{(t)}, \boldsymbol{v}_d \rangle^3 - \beta_e^3 \langle \boldsymbol{w}_j^{(t)}, \boldsymbol{v}_e \rangle^3 - y_i \langle \boldsymbol{w}_j^{(t)}, \boldsymbol{\xi}_i \rangle^3 \right) \tag{12}$$

while for $\alpha N + 1 \leq i \leq N$, $l_i^{(t)}$ can be rewritten as

$$l_i^{(t)} = \text{sigmoid} \left( \sum_{j=1}^{J} -\beta_d^3 \langle \boldsymbol{w}_j^{(t)}, \boldsymbol{v}_d \rangle^3 - y_i \langle \boldsymbol{w}_j^{(t)}, \boldsymbol{\xi}_i \rangle^3 \right)$$

$$\geq \text{sigmoid} \left( \sum_{j=1}^{J} -\beta_d^3 \langle \boldsymbol{w}_j^{(t)}, \boldsymbol{v}_d \rangle^3 - \beta_e^3 \langle \boldsymbol{w}_j^{(t)}, \boldsymbol{v}_e \rangle^3 - y_i \langle \boldsymbol{w}_j^{(t)}, \boldsymbol{\xi}_i \rangle^3 \right) \tag{13}$$

Note that $0 < l_i^{(t)} < 1$ due to the property of the sigmoid function. Furthermore, we similarly consider that the sum of the sigmoid terms for all time steps is bounded up to a logarithmic dependence [11]. The sigmoid term is considered small for a $\kappa$ such that

$$\sum_{t=0}^{T} \frac{1}{1 + \exp(\kappa)} \leq \tilde{O}(1), \tag{14}$$

which implies $\kappa \geq \tilde{\Omega}(1)$.

We present the detailed proofs that build up to Theorem A.8. We begin by considering the update for the fast-learnable and slow-learnable features.

**Lemma A.3** (Fast-learnable feature update.). *For all $t \geq 0$ and $j \in [J]$, the fast-learnable feature update is*

$$\langle \boldsymbol{w}_j^{(t+1)}, \boldsymbol{v}_e \rangle = \langle \boldsymbol{w}_j^{(t)}, \boldsymbol{v}_e \rangle + \tilde{\Theta}(\eta) \alpha \beta_e^3 g_1(t) \langle \boldsymbol{w}_j^{(t)}, \boldsymbol{v}_e \rangle^2, \tag{15}$$

*where $g_1(t) = \text{sigmoid} \left( \sum_{j=1}^{J} -\beta_d^3 \langle \boldsymbol{w}_j^{(t)}, \boldsymbol{v}_d \rangle^3 - \beta_e^3 \langle \boldsymbol{w}_j^{(t)}, \boldsymbol{v}_e \rangle^3 \right)$.*

*Proof.* Plugging the update rule of GD, we have

$$\langle \boldsymbol{w}_j^{(t+1)}, \boldsymbol{v}_e \rangle = \langle \boldsymbol{w}_j^{(t)} - \eta \nabla_{\boldsymbol{w}_j^{(t)}} \mathcal{L}(\boldsymbol{W}^{(t)}), \boldsymbol{v}_e \rangle$$

$$= \langle \boldsymbol{w}_j^{(t)}, \boldsymbol{v}_e \rangle + \frac{3\eta\beta_e^3}{N} \sum_{i=1}^{\alpha N} l_i^{(t)} \langle \boldsymbol{w}_j^{(t)}, \boldsymbol{v}_e \rangle^2$$

$$= \langle \boldsymbol{w}_j^{(t)}, \boldsymbol{v}_e \rangle + \tilde{\Theta}(\eta) \alpha \beta_e^3 g_1(t) \langle \boldsymbol{w}_j^{(t)}, \boldsymbol{v}_e \rangle^2,$$

where the last equality holds due to Lemma B.5. □

Similarly, we obtain the following update rule for slow-learnable features.

**Lemma A.4** (Slow-learnable feature update.)**.** *For all $t \geq 0$ and $j \in [J]$, the fast-learnable feature update is*

$$\langle \boldsymbol{w}_j^{(t+1)}, \boldsymbol{v}_d \rangle = \langle \boldsymbol{w}_j^{(t)}, \boldsymbol{v}_d \rangle + \frac{3\eta\beta_d^3}{N} \sum_{i=1}^{N} l_i^{(t)} \langle \boldsymbol{w}_j^{(t)}, \boldsymbol{v}_d \rangle^2, \tag{16}$$

*which gives*

$$\tilde{\Theta}(\eta)\beta_d^3 g_1(t)\langle \boldsymbol{w}_j^{(t)}, \boldsymbol{v}_d \rangle^2 \leq \langle \boldsymbol{w}_j^{(t+1)}, \boldsymbol{v}_d \rangle - \langle \boldsymbol{w}_j^{(t)}, \boldsymbol{v}_d \rangle \leq \tilde{\Theta}(\eta)\beta_d^3(\alpha g_1(t) + 1 - \alpha)\langle \boldsymbol{w}_j^{(t)}, \boldsymbol{v}_d \rangle^2 \tag{17}$$

*where $g_1(t) = sigmoid\left(\sum_{j=1}^{J} -\beta_d^3\langle \boldsymbol{w}_j^{(t)}, \boldsymbol{v}_d \rangle^3 - \beta_e^3\langle \boldsymbol{w}_j^{(t)}, \boldsymbol{v}_e \rangle^3\right).$*

*Proof.* Plugging the update rule of GD, we have

$$\langle \boldsymbol{w}_j^{(t+1)}, \boldsymbol{v}_d \rangle = \langle \boldsymbol{w}_j^{(t)} - \eta\nabla_{\boldsymbol{w}_j^{(t)}}\mathcal{L}(\boldsymbol{W}^{(t)}), \boldsymbol{v}_d \rangle$$

$$= \langle \boldsymbol{w}_j^{(t)}, \boldsymbol{v}_d \rangle + \frac{3\eta\beta_d^3}{N} \sum_{i=1}^{N} l_i^{(t)} \langle \boldsymbol{w}_j^{(t)}, \boldsymbol{v}_d \rangle^2$$

From Lemma B.5, we have for $1 \leq i \leq \alpha N, l_i^{(t)} = \Theta(1)g_1(t)$ and for $\alpha N + 1 \leq i \leq N, \Theta(1)g_1(t) \leq l_i^{(t)} \leq 1$. Combining with the above equality, we obtain the desired inequalities. $\qquad\square$

Next, we simplify the two above update rules in the early training stage.

**Lemma A.5** (Fast-learnable feature update in early iterations)**.** *Let $T_0 > 0$ be such that $\max_{j \in [J]}\langle \boldsymbol{w}_j^{(T_0)}, \boldsymbol{v}_e \rangle \geq \tilde{\Omega}(1/\beta_e)$. For $t \in [0, T_0]$, the fast-learnable feature update has the following rule*

$$\langle \boldsymbol{w}_j^{(t+1)}, \boldsymbol{v}_e \rangle = \langle \boldsymbol{w}_j^{(t)}, \boldsymbol{v}_e \rangle + \tilde{\Theta}(\eta)\alpha\beta_e^3\langle \boldsymbol{w}_j^{(t)}, \boldsymbol{v}_e \rangle^2, \tag{18}$$

*Proof.* Let $T_0 > 0$ be such that either $\max_{j \in [J]}\langle \boldsymbol{w}_j^{(T_0)}, \boldsymbol{v}_e \rangle \geq \tilde{\Omega}(1/\beta_e)$ or $\max_{j \in [J]}\langle \boldsymbol{w}_j^{(T_0)}, \boldsymbol{v}_d \rangle \geq \tilde{\Omega}(1/\beta_d)$. We will show later that the first condition will be met and we have $\max_{j \in [J]}\langle \boldsymbol{w}_j^{(T_0)}, \boldsymbol{v}_d \rangle \leq \tilde{\Omega}(1/\beta_d)$ for all $j \in [J]$ and $t \in [0, T_0]$.

Recall that $g_1(t) = sigmoid\left(\sum_{j=1}^{J} -\beta_d^3\langle \boldsymbol{w}_j^{(t)}, \boldsymbol{v}_d \rangle^3 - \beta_e^3\langle \boldsymbol{w}_j^{(t)}, \boldsymbol{v}_e \rangle^3\right)$. Then, for $t \in [0, T_0]$, we have

$$g_1(t) = \frac{1}{1 + \exp(\sum_{j=1}^{J} -\beta_d^3\langle \boldsymbol{w}_j^{(t)}, \boldsymbol{v}_d \rangle^3 - \beta_e^3\langle \boldsymbol{w}_j^{(t)}, \boldsymbol{v}_e \rangle^3)}$$

$$\geq \frac{1}{1 + \exp(\kappa + \kappa)}$$

$$= \frac{1}{1 + \exp(\tilde{\Omega}(1))},$$

where the first inequality holds due to $\langle \boldsymbol{w}_j^{(t)}, \boldsymbol{v}_e \rangle \leq \kappa/(J^{1/3}\beta_e)$ and $\langle \boldsymbol{w}_j^{(t)}, \boldsymbol{v}_d \rangle \leq \kappa/(J^{1/3}\beta_d)$ for $t \in [0, T_0]$ [18][Lemma E.3]. Therefore, similar to [18, 32], we have $g_1(t) = \Theta(1)$ in the early iterations. This implies the result in Lemma A.3 as

$$\langle \boldsymbol{w}_j^{(t+1)}, \boldsymbol{v}_e \rangle = \langle \boldsymbol{w}_j^{(t)}, \boldsymbol{v}_e \rangle + \tilde{\Theta}(\eta)\alpha\beta_e^3\langle \boldsymbol{w}_j^{(t)}, \boldsymbol{v}_e \rangle^2. \tag{19}$$

$\square$

Similarly, we obtain the following simplified update rule for slow-learnable features in the early iterations.

**Lemma A.6** (Slow-learnable feature update in early iterations)**.** *Let $T_0 > 0$ be such that $\max_{j \in [J]}\langle \boldsymbol{w}_j^{(T_0)}, \boldsymbol{v}_e \rangle \geq \tilde{\Omega}(1/\beta_e)$. For $t \in [0, T_0]$, the fast-learnable feature update has the following rule*

$$\langle \boldsymbol{w}_j^{(t+1)}, \boldsymbol{v}_d \rangle = \langle \boldsymbol{w}_j^{(t)}, \boldsymbol{v}_d \rangle + \tilde{\Theta}(\eta)\beta_d^3\langle \boldsymbol{w}_j^{(t)}, \boldsymbol{v}_d \rangle^2, \tag{20}$$

We next show that GD will learn the fast-learnable feature quicker than learning the slow-learnable feature.

**Lemma A.7.** *Assume $\eta = \tilde{o}(\beta_d \sigma_0)$. Let $T_0$ be the iteration number that $\max_{j \in [J]} \langle \boldsymbol{w}_j^{(T_0)}, \boldsymbol{v}_e \rangle$ reaches $\tilde{\Omega}(1/\beta_e) = \tilde{\Theta}(1)$. Then, we have for all $t \leq T_0$, it holds that $\max_{j \in [J]} \langle \boldsymbol{w}_j^{(T_0)}, \boldsymbol{v}_d \rangle = \tilde{O}(\sigma_0)$.*

*Proof.* Among all the possible indices $j \in [J]$, we focus on the index $j^\star = \arg \max_{j \in [J]} \langle \boldsymbol{w}_j^{(0)}, \boldsymbol{v}_e \rangle$. Therefore, for $C_t = \alpha \beta_e^3 = \Theta(1)$, we apply Lemma B.2 with two positive sequences $\langle \boldsymbol{w}_{j^\star}^{(t)}, \boldsymbol{v}_e \rangle$ and $\langle \boldsymbol{w}_j^{(t)}, \boldsymbol{v}_d \rangle$ defined in Lemmas A.5 and A.6 and get

$$\langle \boldsymbol{w}_j^{(t)}, \boldsymbol{v}_d \rangle \leq O(\langle \boldsymbol{w}_j^{(0)}, \boldsymbol{v}_d \rangle) = \tilde{O}(\sigma_0) \tag{21}$$

for all $j \in [J]$. $\square$

**Theorem A.8** (Restatement of Theorem 3.2). *We consider training a two-layer nonlinear CNN model initialized with $\boldsymbol{W}^{(0)} \sim \mathcal{N}(0, \sigma_0^2)$ on the training dataset $D = \{(\boldsymbol{x}_i, y_i)\}_{i=1}^N$ that follows the data distribution $\mathcal{D}(\beta_e, \beta_d, \alpha)$ with $\alpha^{1/3} \beta_e > \beta_d$. After training with GD in Equation 3 for $T_{GD}$ iterations where*

$$T_{GD} = \frac{\tilde{\Theta}(1)}{\eta \alpha \beta_e^3 \sigma_0} + \tilde{\Theta}(1) \left\lceil \frac{-\log(\sigma_0 \beta_e)}{\log(2)} \right\rceil, \tag{22}$$

*for all $j \in [J]$ and $t \in [0, T_{GD})$, we have*

$$\langle \boldsymbol{w}_j^{(t+1)}, \boldsymbol{v}_e \rangle = \langle \boldsymbol{w}_j^{(t)}, \boldsymbol{v}_e \rangle + \tilde{\Theta}(\eta) \alpha \beta_e^3 \langle \boldsymbol{w}_j^{(t)}, \boldsymbol{v}_e \rangle^2. \tag{23}$$

$$\langle \boldsymbol{w}_j^{(t+1)}, \boldsymbol{v}_d \rangle = \langle \boldsymbol{w}_j^{(t)}, \boldsymbol{v}_d \rangle + \tilde{\Theta}(\eta) \beta_d^3 \langle \boldsymbol{w}_j^{(t)}, \boldsymbol{v}_d \rangle^2 \tag{24}$$

*After training for $T_{GD}$ iterations, with high probability, the learned weight has the following properties: (1) it learns the fast-learnable feature $\boldsymbol{v}_e$ : $\max_{j \in [J]} \langle \boldsymbol{w}_j^{(T_{GD})}, \boldsymbol{v}_e \rangle \geq \tilde{\Omega}(1/\beta_e)$; (2) it does not learn the slow-learnable feature $\boldsymbol{v}_d$ : $\max_{j \in [J]} \langle \boldsymbol{w}_j^{(T_{GD})}, \boldsymbol{v}_d \rangle = \tilde{O}(\sigma_0)$.*

*Proof.* From the results of Lemmas A.5- A.7, it remains to calculate the time $T_{GD}$. Plugging $v = \tilde{\Omega}(1/\beta_e), m = M = \tilde{\Theta}(\eta) \alpha \beta_e^3, z_0 = \tilde{O}(\sigma_0)$ into Lemma B.3, we have $T_{GD}$ as

$$T_{GD} = \frac{\tilde{\Theta}(1)}{\eta \alpha \beta_e^3 \sigma_0} + \tilde{\Theta}(1) \left\lceil \frac{-\log(\sigma_0 \beta_e)}{\log(2)} \right\rceil \tag{25}$$

$\square$

## A.2   Proof of Theorem 3.3

Before going into the analysis, we denote the derivative of a data example $i$ at iteration $t$ to be

$$l_{i,\boldsymbol{\epsilon}}^{(t)} = \frac{\exp(-y_i f(\boldsymbol{x}_i; \boldsymbol{W}^{(t)} + \boldsymbol{\epsilon}^{(t)}))}{1 + \exp(-y_i f(\boldsymbol{x}_i; \boldsymbol{W}^{(t)} + \boldsymbol{\epsilon}^{(t)}))} = \text{sigmoid}(-y_i f(\boldsymbol{x}_i; \boldsymbol{W}^{(t)} + \boldsymbol{\epsilon}^{(t)})), \tag{26}$$

where $\boldsymbol{\epsilon}^{(t)} = \rho^{(t)} \nabla \mathcal{L}(\boldsymbol{W}^{(t)})$ is the weighted ascent direction at the current parameter $\boldsymbol{W}^{(t)}$. We denote the weight vector of the $j$-th filter after being perturbed by SAM as

$$\boldsymbol{w}_{j,\boldsymbol{\epsilon}}^{(t)} = \boldsymbol{w}_j^{(t)} + \boldsymbol{\epsilon}_j^{(t)} = \boldsymbol{w}_j^{(t)} + \rho^{(t)} \nabla_{\boldsymbol{w}_j^{(t)}} \mathcal{L}(\boldsymbol{W}^{(t)}), \tag{27}$$

where $\rho^{(t)} = \rho / \left\| \nabla \mathcal{L}(\boldsymbol{W}^{(t)}) \right\|_F$.

First, we have the following inequalities regarding the gradient norm:

$$\left\| \nabla \mathcal{L}(\boldsymbol{W}^{(t)}) \right\|_F \geq \left\| \nabla_{\boldsymbol{w}_j^{(t)}} \mathcal{L}(\boldsymbol{W}^{(t)}) \right\| \tag{28}$$

$$= \langle \nabla_{\boldsymbol{w}_j^{(t)}} \mathcal{L}(\boldsymbol{W}^{(t)}), \nabla_{\boldsymbol{w}_j^{(t)}} \mathcal{L}(\boldsymbol{W}^{(t)}) \rangle^{1/2} \tag{29}$$

$$= \left[ \left( \frac{3\beta_d^3}{N} \sum_{i=1}^N l_i^{(t)} \langle \boldsymbol{w}_j^{(t)}, \boldsymbol{v}_d \rangle^2 \right)^2 + \left( \frac{3\beta_e^3}{N} \sum_{i=1}^{\alpha N} l_i^{(t)} \langle \boldsymbol{w}_j^{(t)}, \boldsymbol{v}_e \rangle^2 \right)^2 \right.$$

$$\left. + \left\| \frac{3}{N} \sum_{i=1}^N l_i^{(t)} y_i \langle \boldsymbol{w}_j^{(t)}, \boldsymbol{\xi}_i \rangle^2 \boldsymbol{\xi}_i \right\| \right]^{1/2} \tag{30}$$

Thus,

$$\left\| \nabla \mathcal{L}(\boldsymbol{W}^{(t)}) \right\|_F \geq \frac{3\beta_d^3}{N} \sum_{i=1}^N l_i^{(t)} \langle \boldsymbol{w}_j^{(t)}, \boldsymbol{v}_d \rangle^2 \tag{31}$$

$$\left\| \nabla \mathcal{L}(\boldsymbol{W}^{(t)}) \right\|_F \geq \frac{3\beta_e^3}{N} \sum_{i=1}^{\alpha N} l_i^{(t)} \langle \boldsymbol{w}_j^{(t)}, \boldsymbol{v}_e \rangle^2 \tag{32}$$

**Lemma A.9** (Gradient). *Let the loss function $\mathcal{L}$ be as defined in Equation 2. For $t \geq 0$ and $j \in [J]$, the gradient of the loss $\mathcal{L}(\boldsymbol{W}^{(t)} + \boldsymbol{\epsilon}^{(t)})$ with regard to neuron $\boldsymbol{w}_{j,\boldsymbol{\epsilon}}^{(t)}$ is*

$$\nabla_{\boldsymbol{w}_{j,\boldsymbol{\epsilon}}^{(t)}} \mathcal{L}(\boldsymbol{W}^{(t)} + \boldsymbol{\epsilon}^{(t)}) = -\frac{3}{N} \sum_{i=1}^{\alpha N} l_{i,\boldsymbol{\epsilon}}^{(t)} \left( \beta_d^3 \langle \boldsymbol{w}_{j,\boldsymbol{\epsilon}}^{(t)}, \boldsymbol{v}_d \rangle^2 \boldsymbol{v}_d + \beta_e^3 \langle \boldsymbol{w}_{j,\boldsymbol{\epsilon}}^{(t)}, \boldsymbol{v}_e \rangle^2 \boldsymbol{v}_e + y_i \langle \boldsymbol{w}_{j,\boldsymbol{\epsilon}}^{(t)}, \boldsymbol{\xi}_i \rangle^2 \boldsymbol{\xi}_i \right) -$$

$$\frac{3}{N} \sum_{i=\alpha N+1}^N l_{i,\boldsymbol{\epsilon}}^{(t)} \left( \beta_d^3 \langle \boldsymbol{w}_{j,\boldsymbol{\epsilon}}^{(t)}, \boldsymbol{v}_d \rangle^2 \boldsymbol{v}_d + y_i \langle \boldsymbol{w}_{j,\boldsymbol{\epsilon}}^{(t)}, \boldsymbol{\xi}_i \rangle^2 \boldsymbol{\xi}_i \right) \tag{33}$$

*Proof.* We have the following gradient

$$\nabla_{\boldsymbol{w}_{j,\boldsymbol{\epsilon}}^{(t)}} \mathcal{L}(\boldsymbol{W}^{(t)} + \boldsymbol{\epsilon}^{(t)}) = -\frac{1}{N} \sum_{i=1}^N \frac{\exp(-y_i f(\boldsymbol{x}_i; \boldsymbol{W}^{(t)} + \boldsymbol{\epsilon}^{(t)}))}{1 + \exp(-y_i f(\boldsymbol{x}_i; \boldsymbol{W}^{(t)} + \boldsymbol{\epsilon}^{(t)}))} \cdot y_i f'(\boldsymbol{x}_i; \boldsymbol{W}^{(t)} + \boldsymbol{\epsilon}^{(t)})$$

$$= -\frac{3}{N} \sum_{i=1}^N l_{i,\boldsymbol{\epsilon}}^{(t)} y_i \sum_{p=1}^P \langle \boldsymbol{w}_{j,\boldsymbol{\epsilon}}^{(t)}, \boldsymbol{x}^{(p)} \rangle^2 \cdot \boldsymbol{x}^{(p)}$$

$$= -\frac{3}{N} \sum_{i=1}^{\alpha N} l_{i,\boldsymbol{\epsilon}}^{(t)} \left( \beta_d^3 \langle \boldsymbol{w}_{j,\boldsymbol{\epsilon}}^{(t)}, \boldsymbol{v}_d \rangle^2 \boldsymbol{v}_d + \beta_e^3 \langle \boldsymbol{w}_{j,\boldsymbol{\epsilon}}^{(t)}, \boldsymbol{v}_e \rangle^2 \boldsymbol{v}_e + y_i \langle \boldsymbol{w}_{j,\boldsymbol{\epsilon}}^{(t)}, \boldsymbol{\xi}_i \rangle^2 \boldsymbol{\xi}_i \right) -$$

$$\frac{3}{N} \sum_{i=\alpha N+1}^N l_{i,\boldsymbol{\epsilon}}^{(t)} \left( \beta_d^3 \langle \boldsymbol{w}_{j,\boldsymbol{\epsilon}}^{(t)}, \boldsymbol{v}_d \rangle^2 \boldsymbol{v}_d + y_i \langle \boldsymbol{w}_{j,\boldsymbol{\epsilon}}^{(t)}, \boldsymbol{\xi}_i \rangle^2 \boldsymbol{\xi}_i \right)$$

$\square$

With the above formula of gradient, we have the projection of perturbed weight on $\boldsymbol{v}_e$ is

$$\langle \boldsymbol{w}_{j,\boldsymbol{\epsilon}}^{(t)}, \boldsymbol{v}_e \rangle = \langle \boldsymbol{w}_j^{(t)}, \boldsymbol{v}_e \rangle + \langle \boldsymbol{\epsilon}_j^{(t)}, \boldsymbol{v}_e \rangle$$

$$= \langle \boldsymbol{w}_j^{(t)}, \boldsymbol{v}_e \rangle + \langle \rho^{(t)} \nabla_{\boldsymbol{w}_j^{(t)}} \mathcal{L}(\boldsymbol{W}^{(t)}), \boldsymbol{v}_e \rangle$$

$$= \langle \boldsymbol{w}_j^{(t)}, \boldsymbol{v}_e \rangle - \frac{3\rho^{(t)} \beta_e^3}{N} \sum_{i=1}^{\alpha N} l_i^{(t)} \langle \boldsymbol{w}_j^{(t)}, \boldsymbol{v}_e \rangle^2 \tag{34}$$

From Equations 32 and 34, we have

$$0 \leq \langle \boldsymbol{w}_j^{(t)}, \boldsymbol{v}_e \rangle - \rho \leq \langle \boldsymbol{w}_{j,\boldsymbol{\epsilon}}^{(t)}, \boldsymbol{v}_e \rangle \leq \langle \boldsymbol{w}_j^{(t)}, \boldsymbol{v}_e \rangle \tag{35}$$

Similarly, the projection of perturbed weight on $\boldsymbol{v}_d$ is

$$\langle \boldsymbol{w}_{j,\boldsymbol{\epsilon}}^{(t)}, \boldsymbol{v}_d \rangle = \langle \boldsymbol{w}_j^{(t)}, \boldsymbol{v}_d \rangle - \frac{3\rho^{(t)}\beta_d^3}{N} \sum_{i=1}^{N} l_i^{(t)} \langle \boldsymbol{w}_j^{(t)}, \boldsymbol{v}_d \rangle^2 \tag{36}$$

$$0 \le \langle \boldsymbol{w}_j^{(t)}, \boldsymbol{v}_d \rangle - \rho \le \langle \boldsymbol{w}_{j,\boldsymbol{\epsilon}}^{(t)}, \boldsymbol{v}_d \rangle \le \langle \boldsymbol{w}_j^{(t)}, \boldsymbol{v}_d \rangle \tag{37}$$

**Fast-learnable feature gradient.** The projection of the gradient on $\boldsymbol{v}_e$ is

$$\langle \nabla_{\boldsymbol{w}_{j,\boldsymbol{\epsilon}}^{(t)}} \mathcal{L}(\boldsymbol{W}^{(t)} + \boldsymbol{\epsilon}^{(t)}), \boldsymbol{v}_e \rangle = -\frac{3\beta_e^3}{N} \sum_{i=1}^{\alpha N} l_{i,\boldsymbol{\epsilon}}^{(t)} \langle \boldsymbol{w}_{j,\boldsymbol{\epsilon}}^{(t)}, \boldsymbol{v}_e \rangle^2 \tag{38}$$

**Slow-learnable feature gradient.** The projection of the gradient on $\boldsymbol{v}_d$ is

$$\langle \nabla_{\boldsymbol{w}_{j,\boldsymbol{\epsilon}}^{(t)}} \mathcal{L}(\boldsymbol{W}^{(t)} + \boldsymbol{\epsilon}^{(t)}), \boldsymbol{v}_d \rangle = -\frac{3\beta_d^3}{N} \sum_{i=1}^{N} l_{i,\boldsymbol{\epsilon}}^{(t)} \langle \boldsymbol{w}_{j,\boldsymbol{\epsilon}}^{(t)}, \boldsymbol{v}_d \rangle^2 \tag{39}$$

**Noise gradient.** The projection of the gradient on $\boldsymbol{\xi}_i$ is

$$\langle \nabla_{\boldsymbol{w}_{j,\boldsymbol{\epsilon}}^{(t)}} \mathcal{L}(\boldsymbol{W}^{(t)} + \boldsymbol{\epsilon}^{(t)}), \boldsymbol{\xi}_i \rangle = -\frac{3}{N} \left( l_{i,\boldsymbol{\epsilon}}^{(t)} y_i \langle \boldsymbol{w}_{j,\boldsymbol{\epsilon}}^{(t)}, \boldsymbol{\xi}_i \rangle^2 \|\boldsymbol{\xi}_i\|_2^2 + \sum_{k=1, k\neq i}^{N} l_{k,\boldsymbol{\epsilon}}^{(t)} y_k \langle \boldsymbol{w}_{j,\boldsymbol{\epsilon}}^{(t)}, \boldsymbol{\xi}_k \rangle^2 \langle \boldsymbol{\xi}_k, \boldsymbol{\xi}_i \rangle \right) \tag{40}$$

**Derivative of data example $i$.** For $1 \le i \le \alpha N$, $l_{i,\boldsymbol{\epsilon}}^{(t)}$ can be rewritten as

$$l_{i,\boldsymbol{\epsilon}}^{(t)} = \text{sigmoid} \left( \sum_{j=1}^{J} -\beta_d^3 \langle \boldsymbol{w}_{j,\boldsymbol{\epsilon}}^{(t)}, \boldsymbol{v}_d \rangle^3 - \beta_e^3 \langle \boldsymbol{w}_{j,\boldsymbol{\epsilon}}^{(t)}, \boldsymbol{v}_e \rangle^3 - y_i \langle \boldsymbol{w}_{j,\boldsymbol{\epsilon}}^{(t)}, \boldsymbol{\xi}_i \rangle^3 \right) \tag{41}$$

while for $\alpha N + 1 \le i \le N$, $l_i^{(t)}$ can be rewritten as

$$l_{i,\boldsymbol{\epsilon}}^{(t)} = \text{sigmoid} \left( \sum_{j=1}^{J} -\beta_d^3 \langle \boldsymbol{w}_{j,\boldsymbol{\epsilon}}^{(t)}, \boldsymbol{v}_d \rangle^3 - y_i \langle \boldsymbol{w}_{j,\boldsymbol{\epsilon}}^{(t)}, \boldsymbol{\xi}_i \rangle^3 \right)$$

$$\ge \text{sigmoid} \left( \sum_{j=1}^{J} -\beta_d^3 \langle \boldsymbol{w}_{j,\boldsymbol{\epsilon}}^{(t)}, \boldsymbol{v}_d \rangle^3 - \beta_e^3 \langle \boldsymbol{w}_{j,\boldsymbol{\epsilon}}^{(t)}, \boldsymbol{v}_e \rangle^3 - y_i \langle \boldsymbol{w}_{j,\boldsymbol{\epsilon}}^{(t)}, \boldsymbol{\xi}_i \rangle^3 \right) \tag{42}$$

We present the detailed proofs that build up to Theorem A.15. We begin by considering the update for the fast-learnable and slow-learnable features.

**Lemma A.10** (Fast-learnable feature update.). *For all $t \ge 0$ and $j \in [J]$, the fast-learnable feature update is*

$$\langle \boldsymbol{w}_j^{(t)}, \boldsymbol{v}_e \rangle + \tilde{\Theta}(\eta)\alpha\beta_e^3 g_2(t)(\langle \boldsymbol{w}_j^{(t)}, \boldsymbol{v}_e \rangle - \rho)^2 \le \langle \boldsymbol{w}_j^{(t+1)}, \boldsymbol{v}_e \rangle = \langle \boldsymbol{w}_j^{(t)}, \boldsymbol{v}_e \rangle + \tilde{\Theta}(\eta)\alpha\beta_e^3 g_2(t)\langle \boldsymbol{w}_{j,\boldsymbol{\epsilon}}^{(t)}, \boldsymbol{v}_e \rangle^2$$

$$\le \langle \boldsymbol{w}_j^{(t)}, \boldsymbol{v}_e \rangle + \tilde{\Theta}(\eta)\alpha\beta_e^3 g_2(t)(\langle \boldsymbol{w}_j^{(t)}, \boldsymbol{v}_e \rangle)^2 \tag{43}$$

*where $g_2(t) = \text{sigmoid} \left( \sum_{j=1}^{J} -\beta_d^3 \langle \boldsymbol{w}_{j,\boldsymbol{\epsilon}}^{(t)}, \boldsymbol{v}_d \rangle^3 - \beta_e^3 \langle \boldsymbol{w}_{j,\boldsymbol{\epsilon}}^{(t)}, \boldsymbol{v}_e \rangle^3 \right)$.*

*Proof.* Plugging the update rule of SAM, we have

$$\langle \boldsymbol{w}_j^{(t+1)}, \boldsymbol{v}_e \rangle = \langle \boldsymbol{w}_j^{(t)} - \eta\nabla_{\boldsymbol{w}_{j,\boldsymbol{\epsilon}}^{(t)}} \mathcal{L}(\boldsymbol{W}^{(t)} + \boldsymbol{\epsilon}^{(t)}), \boldsymbol{v}_e \rangle$$

$$= \langle \boldsymbol{w}_j^{(t)}, \boldsymbol{v}_e \rangle + \frac{3\eta\alpha\beta_e^3}{N} \sum_{i=1}^{N} l_{i,\boldsymbol{\epsilon}}^{(t)} \langle \boldsymbol{w}_{j,\boldsymbol{\epsilon}}^{(t)}, \boldsymbol{v}_e \rangle^2$$

$$= \langle \boldsymbol{w}_j^{(t)}, \boldsymbol{v}_e \rangle + \tilde{\Theta}(\eta)\alpha\beta_e^3 g_2(t)\langle \boldsymbol{w}_{j,\boldsymbol{\epsilon}}^{(t)}, \boldsymbol{v}_e \rangle^2,$$

where the last equality holds due to Lemma B.10. Combining with Equation 35, we obtain the desired inequalities. □

Similarly, we obtain the following update rule for slow-learnable features.

**Lemma A.11** (Slow-learnable feature update.). *For all $t \geq 0$ and $j \in [J]$, the slow-learnable feature update is*

$$\langle \boldsymbol{w}_j^{(t)}, \boldsymbol{v}_d \rangle + \tilde{\Theta}(\eta)\beta_d^3 g_2(t)(\langle \boldsymbol{w}_j^{(t)}, \boldsymbol{v}_d \rangle - \rho)^2 \leq \langle \boldsymbol{w}_j^{(t+1)}, \boldsymbol{v}_d \rangle = \langle \boldsymbol{w}_j^{(t)}, \boldsymbol{v}_d \rangle + \frac{3\eta\beta_d^3}{N} \sum_{i=1}^{N} l_i^{(t)} \langle \boldsymbol{w}_{j,\boldsymbol{\epsilon}}^{(t)}, \boldsymbol{v}_d \rangle^2$$

$$\leq \langle \boldsymbol{w}_j^{(t)}, \boldsymbol{v}_d \rangle + \tilde{\Theta}(\eta)\beta_d^3(\alpha g_2(t) + 1 - \alpha)(\langle \boldsymbol{w}_j^{(t)}, \boldsymbol{v}_d \rangle)^2$$

(44)

*where $g_2(t) = sigmoid\left( \sum_{j=1}^{J} -\beta_d^3 \langle \boldsymbol{w}_{j,\boldsymbol{\epsilon}}^{(t)}, \boldsymbol{v}_d \rangle^3 - \beta_e^3 \langle \boldsymbol{w}_{j,\boldsymbol{\epsilon}}^{(t)}, \boldsymbol{v}_e \rangle^3 \right)$.*

*Proof.* Plugging the update rule of GD, we have

$$\langle \boldsymbol{w}_j^{(t+1)}, \boldsymbol{v}_d \rangle = \langle \boldsymbol{w}_j^{(t)} - \eta \nabla_{\boldsymbol{w}_{j,\boldsymbol{\epsilon}}^{(t)}} \mathcal{L}(\boldsymbol{W}^{(t)}), \boldsymbol{v}_d \rangle$$

$$= \langle \boldsymbol{w}_j^{(t)}, \boldsymbol{v}_d \rangle + \frac{3\eta\beta_d^3}{N} \sum_{i=1}^{N} l_i^{(t)} \langle \boldsymbol{w}_{j,\boldsymbol{\epsilon}}^{(t)}, \boldsymbol{v}_d \rangle^2$$

From Lemma B.5, we have for $1 \leq i \leq \alpha N$, $l_i^{(t)} = \Theta(1)g_1(t)$ and for $\alpha N + 1 \leq i \leq N$, $\Theta(1)g_1(t) \leq l_i^{(t)} \leq 1$. Combining with the above equality and Equation 37, we obtain the desired inequalities. $\square$

Next, we simplify the two above update rules in the early training stage.

**Lemma A.12** (Fast-learnable feature update in early iterations). *Let $T_0 > 0$ be such that $\max_{j \in [J]} \langle \boldsymbol{w}_j^{(T_0)}, \boldsymbol{v}_e \rangle \geq \tilde{\Omega}(1/\beta_e)$. For $t \in [0, T_0]$, the fast-learnable feature update has the following rule*

$$\tilde{\Theta}(\eta)\alpha\beta_e^3(\langle \boldsymbol{w}_j^{(t)}, \boldsymbol{v}_e \rangle - \rho)^2 \leq \langle \boldsymbol{w}_j^{(t+1)}, \boldsymbol{v}_e \rangle - \langle \boldsymbol{w}_j^{(t)}, \boldsymbol{v}_e \rangle \leq \langle \boldsymbol{w}_j^{(t)}, \boldsymbol{v}_e \rangle + \tilde{\Theta}(\eta)\alpha\beta_e^3(\langle \boldsymbol{w}_j^{(t)}, \boldsymbol{v}_e \rangle)^2$$

(45)

*Proof.* Let $T_0 > 0$ be such that either $\max_{j \in [J]} \langle \boldsymbol{w}_j^{(T_0)}, \boldsymbol{v}_e \rangle \geq \tilde{\Omega}(1/\beta_e)$ or $\max_{j \in [J]} \langle \boldsymbol{w}_j^{(T_0)}, \boldsymbol{v}_d \rangle \geq \tilde{\Omega}(1/\beta_d)$. We will show later that the first condition will be met and we have $\max_{j \in [J]} \langle \boldsymbol{w}_j^{(T_0)}, \boldsymbol{v}_d \rangle \leq \tilde{\Omega}(1/\beta_d)$ for all $j \in [J]$ and $t \in [0, T_0]$.

Recall that $g_2(t) = sigmoid\left( \sum_{j=1}^{J} -\beta_d^3 \langle \boldsymbol{w}_{j,\boldsymbol{\epsilon}}^{(t)}, \boldsymbol{v}_d \rangle^3 - \beta_e^3 \langle \boldsymbol{w}_{j,\boldsymbol{\epsilon}}^{(t)}, \boldsymbol{v}_e \rangle^3 \right)$. Then, for $t \in [0, T_0]$, we have

$$g_2(t) = \frac{1}{1 + \exp(\sum_{j=1}^{J} -\beta_d^3 \langle \boldsymbol{w}_{j,\boldsymbol{\epsilon}}^{(t)}, \boldsymbol{v}_d \rangle^3 - \beta_e^3 \langle \boldsymbol{w}_{j,\boldsymbol{\epsilon}}^{(t)}, \boldsymbol{v}_e \rangle^3)}$$

$$\geq \frac{1}{1 + \exp(\kappa + \kappa)}$$

$$= \frac{1}{1 + \exp(\tilde{\Omega}(1))},$$

where the first inequality holds due to $\langle \boldsymbol{w}_{j,\boldsymbol{\epsilon}}^{(t)}, \boldsymbol{v}_e \rangle \leq \langle \boldsymbol{w}_j^{(t)}, \boldsymbol{v}_e \rangle \leq \kappa/(J^{1/3}\beta_e)$ and $\langle \boldsymbol{w}_{j,\boldsymbol{\epsilon}}^{(t)}, \boldsymbol{v}_d \rangle \leq \langle \boldsymbol{w}_j^{(t)}, \boldsymbol{v}_d \rangle \leq \kappa/(J^{1/3}\beta_d)$ for $t \in [0, T_0]$. Therefore, we have $g_2(t) = \Theta(1)$ in the early iterations. Replacing $g_2(t) = \Theta(1)$ into the results of Lemma A.10, we obtain the desired results. $\square$

Similarly, we obtain the following simplified update rule for slow-learnable features in the early iterations.

**Lemma A.13** (Slow-learnable feature update in early iterations). *Let $T_0 > 0$ be such that $\max_{j \in [J]} \langle \boldsymbol{w}_j^{(T_0)}, \boldsymbol{v}_e \rangle \geq \tilde{\Omega}(1/\beta_e)$. For $t \in [0, T_0]$, the fast-learnable feature update has the following rule*

$$\tilde{\Theta}(\eta)\beta_d^3(\langle \boldsymbol{w}_j^{(t)}, \boldsymbol{v}_d \rangle - \rho)^2 \leq \langle \boldsymbol{w}_j^{(t+1)}, \boldsymbol{v}_d \rangle - \langle \boldsymbol{w}_j^{(t)}, \boldsymbol{v}_d \rangle \leq \tilde{\Theta}(\eta)\beta_d^3(\langle \boldsymbol{w}_j^{(t)}, \boldsymbol{v}_d \rangle)^2 \quad (46)$$

We next show that SAM will learn the fast-learnable feature quicker than the slow-learnable one.

**Lemma A.14.** *Assume $\eta = \tilde{o}(\beta_d \sigma_0)$. Let $T_0$ be the iteration number that $\max_{j \in [J]} \langle \boldsymbol{w}_j^{(T_0)}, \boldsymbol{v}_e \rangle$ reaches $\tilde{\Omega}(1/\beta_e) = \tilde{\Theta}(1)$. Then, we have for all $t \leq T_0$, it holds that $\max_{j \in [J]} \langle \boldsymbol{w}_j^{(T_0)}, \boldsymbol{v}_d \rangle = \tilde{O}(\sigma_0)$.*

*Proof.* Among all the possible indices $j \in [J]$, we focus on the index $j^\star = \arg\max_{j \in [J]} \langle \boldsymbol{w}_j^{(0)}, \boldsymbol{v}_e \rangle$. Therefore, for $C_t = \alpha \beta_e^3 = \Theta(1)$, we apply Lemma B.2 with two positive sequences $\langle \boldsymbol{w}_{j^\star}^{(t)}, \boldsymbol{v}_e \rangle$ and $\langle \boldsymbol{w}_j^{(t)}, \boldsymbol{v}_d \rangle$ defined in Lemmas A.12 and A.13 and get

$$\langle \boldsymbol{w}_j^{(t)}, \boldsymbol{v}_d \rangle \leq O(\langle \boldsymbol{w}_j^{(0)}, \boldsymbol{v}_d \rangle) = \tilde{O}(\sigma_0) \tag{47}$$

for all $j \in [J]$. $\square$

**Theorem A.15** (Restatement of Theorem 3.3). *We consider training a two-layer nonlinear CNN model initialized with $\boldsymbol{W}^{(0)} \sim \mathcal{N}(0, \sigma_0^2)$ on the training dataset $D = \{(\boldsymbol{x}_i, y_i)\}_{i=1}^N$ that follows the data distribution $\mathcal{D}(\beta_e, \beta_d, \alpha)$ with $\alpha^{1/3}\beta_e > \beta_d$. After training with SAM in Equation 3 for $T_{SAM}$ iterations where*

$$T_{SAM} = \frac{\tilde{\Theta}(\sigma_0)}{\eta \alpha \beta_e^3 (\sigma_0 - \rho)^2} + \frac{\tilde{\Theta}(\sigma_0^2)}{(\sigma_0 - \rho)^2} \left\lceil \frac{-\log(\sigma_0 \beta_e)}{\log(2)} \right\rceil, \tag{48}$$

*for all $j \in [J]$ and $t \in [0, T_{SAM})$, we have*

$$\langle \boldsymbol{w}_j^{(t)}, \boldsymbol{v}_e \rangle + \tilde{\Theta}(\eta)\alpha\beta_e^3(\langle \boldsymbol{w}_j^{(t)}, \boldsymbol{v}_e \rangle - \rho)^2 \leq \langle \boldsymbol{w}_j^{(t+1)}, \boldsymbol{v}_e \rangle = \langle \boldsymbol{w}_j^{(t)}, \boldsymbol{v}_e \rangle + \tilde{\Theta}(\eta)\alpha\beta_e^3 \langle \boldsymbol{w}_{j,\boldsymbol{\epsilon}}^{(t)}, \boldsymbol{v}_e \rangle^2$$
$$\leq \langle \boldsymbol{w}_j^{(t)}, \boldsymbol{v}_e \rangle + \tilde{\Theta}(\eta)\alpha\beta_e^3(\langle \boldsymbol{w}_j^{(t)}, \boldsymbol{v}_e \rangle)^2 \tag{49}$$

$$\langle \boldsymbol{w}_j^{(t)}, \boldsymbol{v}_d \rangle + \tilde{\Theta}(\eta)\beta_d^3(\langle \boldsymbol{w}_j^{(t)}, \boldsymbol{v}_d \rangle - \rho)^2 \leq \langle \boldsymbol{w}_j^{(t+1)}, \boldsymbol{v}_d \rangle = \langle \boldsymbol{w}_j^{(t)}, \boldsymbol{v}_d \rangle + \tilde{\Theta}(\eta)\beta_d^3 \langle \boldsymbol{w}_{j,\boldsymbol{\epsilon}}^{(t)}, \boldsymbol{v}_d \rangle^2$$
$$\leq \langle \boldsymbol{w}_j^{(t)}, \boldsymbol{v}_d \rangle + \tilde{\Theta}(\eta)\beta_d^3(\langle \boldsymbol{w}_j^{(t)}, \boldsymbol{v}_d \rangle)^2 \tag{50}$$

*After training for $T_{SAM}$ iterations, with high probability, the learned weight has the following properties: (1) it learns the fast-learnable feature $\boldsymbol{v}_e$ : $\max_{j \in [J]} \langle \boldsymbol{w}_j^{(T_{SAM})}, \boldsymbol{v}_e \rangle \geq \tilde{\Omega}(1/\beta_e)$; (2) it does not learn the slow-learnable feature $\boldsymbol{v}_d$ : $\max_{j \in [J]} \langle \boldsymbol{w}_j^{(T_{SAM})}, \boldsymbol{v}_d \rangle = \tilde{O}(\sigma_0)$.*

*Proof.* With the results of Lemmas A.12- A.14, it remains to calculate the time $T_{SAM}$. Plugging $v = \tilde{\Omega}(1/\beta_e), m = M = \tilde{\Theta}(\eta)\alpha\beta_e^3, z_0 = \tilde{O}(\sigma_0)$ into Lemma B.3, we have $T_{SAM}$ as

$$T_{SAM} = \frac{\tilde{\Theta}(\sigma_0)}{\eta \alpha \beta_e^3 (\sigma_0 - \rho)^2} + \frac{\tilde{\Theta}(\sigma_0^2)}{(\sigma_0 - \rho)^2} \left\lceil \frac{-\log(\sigma_0 \beta_e)}{\log(2)} \right\rceil \tag{51}$$

Comparing Eq. 22 and 48, we can see that SAM learns the fast-learnable features later than GD. Particularly, if we remove the approximate notations, we have the following inequality

$$\frac{1}{\eta \beta_e^3 \sigma_0} + \left\lceil \frac{-\log(\sigma_0 \beta_e)}{\log(2)} \right\rceil \geq \frac{\sigma_0}{\eta \beta_e^3(\sigma_0 - \rho)^2} + \frac{\sigma_0^2}{(\sigma_0 - \rho)^2} \left\lceil \frac{-\log(\sigma_0 \beta_e)}{\log(2)} \right\rceil, \tag{52}$$

which holds due to Assumption A.1 about weight initialization in Appendix A.1, i.e., $(\sigma_0 \geq \rho \geq 0)$. $\square$

### A.3 Proof of Theorem 3.4

In this section, we show that SAM learns fast-learnable and slow-learnable features at a more uniform speed. To ease the notation, we denote $G_e^{(t)} = \max_{j \in [J]} \langle \boldsymbol{w}_j^{(t)}, \boldsymbol{v}_e \rangle$ and $G_d^{(t)} = \max_{j \in [J]} \langle \boldsymbol{w}_j^{(t)}, \boldsymbol{v}_d \rangle$ for model weights trained with GD. Similarly, we denote $S_e^{(t)}$ and $S_d^{(t)}$ for model weights trained with SAM. We use $\hat{S}_e^{(t)}$ and $\hat{S}_d^{(t)}$ to denote the inner products with perturbed weights. We simplify Equation 34 and 36 for early iterations $t \leq T_0$ as

$$\hat{S}_e^{(t)} = S_e^{(t)} - \tilde{\Theta}(1)\rho^{(t)}\alpha\beta_e^3(S_e^{(t)})^2 \tag{53}$$
$$\hat{S}_d^{(t)} = S_d^{(t)} - \tilde{\Theta}(1)\rho^{(t)}\beta_d^3(S_d^{(t)})^2 \tag{54}$$

Before introducing the theorem, we assume that the model is initialized in favor of the fast-learnable feature, i.e. $G_e^{(0)} - G_d^{(0)} \geq \rho$. This is reasonable as a consequence of Theorem A.8 because we can just train the model for several iterations to achieve this initialization (similar argument for Assumption A.1).

**Theorem A.16** (Restatement of Theorem 3.4). *Consider the training dataset $D = \{(\boldsymbol{x}_i, y_i)\}_{i=1}^N$ that follows the data distribution $\mathcal{D}(\beta_e, \beta_d, \alpha)$ using the two-layer nonlinear CNN model initialized with $\boldsymbol{W}^{(0)} \sim \mathcal{N}(0, \sigma_0^2)$. Assume that the fast-learnable feature strength is significantly larger $\alpha^{1/3}\beta_e > \beta_d$. Training the same model initialization, we have that for every iteration $t \leq T_0$*

$$\rho + S_d^{(t)} \leq S_e^{(t)} \tag{55}$$

$$\hat{S}_d^{(t)} < \hat{S}_e^{(t)} \tag{56}$$

$$S_e^{(t)} \leq G_e^{(t)} \tag{57}$$

$$S_d^{(t)} \leq G_d^{(t)} \tag{58}$$

$$S_e^{(t)} - S_d^{(t)} \leq G_e^{(t)} - G_d^{(t)} \tag{59}$$

*Proof.* We prove this by induction. For $t = 0$, the above hypotheses immediately hold because we use train two methods from the same initialization. Particularly, we have $0 < S_d^{(0)} = G_d^{(0)} < G_e^{(0)} = S_e^{(0)}$ and $\hat{S}_e^{(0)} - \hat{S}_d^{(0)} \geq S_e^{(0)} - \rho - S_d^{(0)} \geq (G_e^{(0)} - G_d^{(0)}) - \rho \geq 0$.

Assume that the induction hypotheses hold for $t$, i.e.

$$\rho + S_d^{(t)} \leq S_e^{(t)} \tag{60}$$

$$\hat{S}_d^{(t)} < \hat{S}_e^{(t)} \tag{61}$$

$$S_e^{(t)} \leq G_e^{(t)} \tag{62}$$

$$S_d^{(t)} \leq G_d^{(t)} \tag{63}$$

$$S_e^{(t)} - S_d^{(t)} \leq G_e^{(t)} - G_d^{(t)} \tag{64}$$

We need to prove that they also hold for $t + 1$. From Lemma A.15 and the first two induction hypotheses,

$$\rho + S_d^{(t+1)} = \rho + S_d^{(t)} + \tilde{\Theta}(\eta)\beta_d^3(\hat{S}_d^{(t)})^2 < S_e^{(t)} + \tilde{\Theta}(\eta)\alpha\beta_e^3(\hat{S}_e^{(t)})^2 = S_e^{(t+1)} \tag{65}$$

Then, $\hat{S}_e^{(t+1)} - \hat{S}_d^{(t+1)} \geq S_e^{(t+1)} - \rho - S_d^{(t+1)} \geq 0$. From Equation 35 and Lemma A.12,

$$S_e^{(t+1)} \leq S_e^{(t)} + \tilde{\Theta}(\eta)\alpha\beta_e^3(S_e^{(t)})^2 \leq G_e^{(t)} + \tilde{\Theta}(\eta)\alpha\beta_e^3(G_e^{(t)})^2 \leq G_e^{(t+1)}. \tag{66}$$

Similarly, we have $S_d^{(t+1)} \leq G_d^{(t+1)}$. From Equations 53 and 54,

$$S_d^{(t)} - \hat{S}_d^{(t)} = \tilde{\Theta}(1)\rho^{(t)}\beta_d^3(S_d^{(t)})^2) < \tilde{\Theta}(1)\rho^{(t)}\alpha\beta_e^3(S_e^{(t)})^2 = S_e^{(t)} - \hat{S}_e^{(t)} \tag{67}$$

$$0 \leq \hat{S}_e^{(t)} - \hat{S}_d^{(t)} < S_e^{(t)} - S_d^{(t)} \leq G_e^{(t)} - G_d^{(t)} \tag{68}$$

Combining with Equations 35 and 37, we have

$$(\hat{S}_e^{(t)})^2 - (\hat{S}_d^{(t)})^2 < (S_e^{(t)})^2 - (S_d^{(t)})^2 \leq (G_e^{(t)})^2 - (G_d^{(t)})^2 \tag{69}$$

$$(G_d^{(t)})^2 - (\hat{S}_d^{(t)})^2 < (G_e^{(t)})^2 - (\hat{S}_e^{(t)})^2 \tag{70}$$

$$\tilde{\Theta}(\eta)\beta_d^3((G_d^{(t)})^2 - (\hat{S}_d^{(t)})^2) < \tilde{\Theta}(\eta)\alpha\beta_e^3((G_e^{(t)})^2 - (\hat{S}_e^{(t)})^2) \tag{71}$$

$$G_d^{(t)} - S_d^{(t)} + \tilde{\Theta}(\eta)\beta_d^3((G_d^{(t)})^2 - (\hat{S}_d^{(t)})^2) < G_e^{(t)} - S_e^{(t)} + \tilde{\Theta}(\eta)\alpha\beta_e^3((G_e^{(t)})^2 - (\hat{S}_e^{(t)})^2) \tag{72}$$

$$G_d^{(t+1)} - S_d^{(t+1)} < G_e^{(t+1)} - S_e^{(t+1)} \tag{73}$$

$$S_e^{(t+1)} - S_d^{(t+1)} < G_e^{(t+1)} - G_d^{(t+1)} \tag{74}$$

Therefore, the induction hypotheses hold for $t + 1$. $\qquad\qquad\square$

## A.4 Proof of Theorem 3.5

From Theorem A.16, we have the following result for switching between SAM and GD during training.

**Lemma A.17.** *Consider the training dataset $D = \{(\boldsymbol{x}_i, y_i)\}_{i=1}^N$ that follows the data distribution $\mathcal{D}(\beta_e, \beta_d, \alpha)$ using the two-layer nonlinear CNN model initialized with $\boldsymbol{W}^{(0)} \sim \mathcal{N}(0, \sigma_0^2)$. Assume that the noise is sufficiently small (ref. Lemmas B.4 and B.9) and the fast-learnable feature strength is significantly larger $\alpha^{1/3}\beta_e > \beta_d$. From any iteration $t$ during early training, the normalized gradient of the one-step SAM update has a larger weight on the slow-learnable feature compared to that of GD.*

*Proof.* First, recall the gradients of GD and SAM are as follows.

$$
\nabla_{\boldsymbol{w}_j^{(t)}} \mathcal{L}(\boldsymbol{W}^{(t)}) = -\frac{3}{N} \sum_{i=1}^{\alpha N} l_i^{(t)} \left( \beta_d^3 \langle \boldsymbol{w}_j^{(t)}, \boldsymbol{v}_d \rangle^2 \boldsymbol{v}_d + \beta_e^3 \langle \boldsymbol{w}_j^{(t)}, \boldsymbol{v}_e \rangle^2 \boldsymbol{v}_e + y_i \langle \boldsymbol{w}_j^{(t)}, \boldsymbol{\xi}_i \rangle^2 \boldsymbol{\xi}_i \right) -
$$
$$
\frac{3}{N} \sum_{i=\alpha N+1}^{N} l_i^{(t)} \left( \beta_d^3 \langle \boldsymbol{w}_j^{(t)}, \boldsymbol{v}_d \rangle^2 \boldsymbol{v}_d + y_i \langle \boldsymbol{w}_j^{(t)}, \boldsymbol{\xi}_i \rangle^2 \boldsymbol{\xi}_i \right) \tag{75}
$$

$$
\nabla_{\boldsymbol{w}_{j,\boldsymbol{\epsilon}}^{(t)}} \mathcal{L}(\boldsymbol{W}^{(t)} + \boldsymbol{\epsilon}^{(t)}) = -\frac{3}{N} \sum_{i=1}^{\alpha N} l_{i,\boldsymbol{\epsilon}}^{(t)} \left( \beta_d^3 \langle \boldsymbol{w}_{j,\boldsymbol{\epsilon}}^{(t)}, \boldsymbol{v}_d \rangle^2 \boldsymbol{v}_d + \beta_e^3 \langle \boldsymbol{w}_{j,\boldsymbol{\epsilon}}^{(t)}, \boldsymbol{v}_e \rangle^2 \boldsymbol{v}_e + y_i \langle \boldsymbol{w}_{j,\boldsymbol{\epsilon}}^{(t)}, \boldsymbol{\xi}_i \rangle^2 \boldsymbol{\xi}_i \right) -
$$
$$
\frac{3}{N} \sum_{i=\alpha N+1}^{N} l_{i,\boldsymbol{\epsilon}}^{(t)} \left( \beta_d^3 \langle \boldsymbol{w}_{j,\boldsymbol{\epsilon}}^{(t)}, \boldsymbol{v}_d \rangle^2 \boldsymbol{v}_d + y_i \langle \boldsymbol{w}_{j,\boldsymbol{\epsilon}}^{(t)}, \boldsymbol{\xi}_i \rangle^2 \boldsymbol{\xi}_i \right) \tag{76}
$$

Because the noise is sufficiently small, the above equations can be simplified as

$$
\nabla_{\boldsymbol{w}^{(t)}} \mathcal{L}(\boldsymbol{W}^{(t)}) = -\left( \frac{3}{N} \sum_{i=1}^{N} l_i^{(t)} \beta_d^3 \langle \boldsymbol{w}^{(t)}, \boldsymbol{v}_d \rangle^2 \right) \boldsymbol{v}_d - \left( \frac{3}{N} \sum_{i=1}^{\alpha N} l_i^{(t)} \beta_e^3 \langle \boldsymbol{w}^{(t)}, \boldsymbol{v}_e \rangle^2 \right) \boldsymbol{v}_e \tag{77}
$$

$$
\nabla_{\boldsymbol{w}_{\boldsymbol{\epsilon}}^{(t)}} \mathcal{L}(\boldsymbol{W}^{(t)} + \boldsymbol{\epsilon}^{(t)}) = -\left( \frac{3}{N} \sum_{i=1}^{N} l_{i,\boldsymbol{\epsilon}}^{(t)} \beta_d^3 \langle \boldsymbol{w}_{\boldsymbol{\epsilon}}^{(t)}, \boldsymbol{v}_d \rangle^2 \right) \boldsymbol{v}_d - \left( \frac{3}{N} \sum_{i=1}^{\alpha N} l_{i,\boldsymbol{\epsilon}}^{(t)} \beta_e^3 \langle \boldsymbol{w}_{\boldsymbol{\epsilon}}^{(t)}, \boldsymbol{v}_e \rangle^2 \right) \boldsymbol{v}_e \tag{78}
$$

Note that in early training, we have an approximation for the logit terms as $l_i^{(t)} = l_{i,\boldsymbol{\epsilon}}^{(t)} = \Theta(1)$, we can further simplify the gradients as

$$
\nabla_{\boldsymbol{w}^{(t)}} \mathcal{L}(\boldsymbol{W}^{(t)}) = -\left( 3\beta_d^3 \langle \boldsymbol{w}^{(t)}, \boldsymbol{v}_d \rangle^2 \right) \boldsymbol{v}_d - \left( 3\alpha \beta_e^3 \langle \boldsymbol{w}^{(t)}, \boldsymbol{v}_e \rangle^2 \right) \boldsymbol{v}_e \tag{79}
$$

$$
\nabla_{\boldsymbol{w}_{\boldsymbol{\epsilon}}^{(t)}} \mathcal{L}(\boldsymbol{W}^{(t)} + \boldsymbol{\epsilon}^{(t)}) = -\left( 3\beta_d^3 \langle \boldsymbol{w}_{\boldsymbol{\epsilon}}^{(t)}, \boldsymbol{v}_d \rangle^2 \right) \boldsymbol{v}_d - \left( 3\alpha \beta_e^3 \langle \boldsymbol{w}_{\boldsymbol{\epsilon}}^{(t)}, \boldsymbol{v}_e \rangle^2 \right) \boldsymbol{v}_e \tag{80}
$$

Both gradients of GD and SAM can be decomposed into the linear combination of fast-learnable and slow-learnable features. To prove that the normalized gradient of SAM favors the slow-learnable feature compared to GD, it is sufficient to show the ratio of coefficients in SAM is larger than GD. In other words, we need to verify that

$$
\frac{3\beta_d^3 \langle \boldsymbol{w}_{\boldsymbol{\epsilon}}^{(t)}, \boldsymbol{v}_d \rangle^2}{3\alpha \beta_e^3 \langle \boldsymbol{w}_{\boldsymbol{\epsilon}}^{(t)}, \boldsymbol{v}_e \rangle^2} \geq \frac{3\beta_d^3 \langle \boldsymbol{w}^{(t)}, \boldsymbol{v}_d \rangle^2}{3\alpha \beta_e^3 \langle \boldsymbol{w}^{(t)}, \boldsymbol{v}_e \rangle^2} \tag{81}
$$

$$
\frac{\langle \boldsymbol{w}_{\boldsymbol{\epsilon}}^{(t)}, \boldsymbol{v}_d \rangle}{\langle \boldsymbol{w}_{\boldsymbol{\epsilon}}^{(t)}, \boldsymbol{v}_e \rangle} \geq \frac{\langle \boldsymbol{w}^{(t)}, \boldsymbol{v}_d \rangle}{\langle \boldsymbol{w}^{(t)}, \boldsymbol{v}_e \rangle} \tag{82}
$$

$$
\frac{\langle \boldsymbol{w}^{(t)}, \boldsymbol{v}_d \rangle - 3\rho^{(t)}\beta_d^3 \langle \boldsymbol{w}^{(t)}, \boldsymbol{v}_d \rangle^2}{\langle \boldsymbol{w}^{(t)}, \boldsymbol{v}_e \rangle - 3\rho^{(t)}\alpha\beta_e^3 \langle \boldsymbol{w}^{(t)}, \boldsymbol{v}_e \rangle^2} \geq \frac{\langle \boldsymbol{w}^{(t)}, \boldsymbol{v}_d \rangle}{\langle \boldsymbol{w}^{(t)}, \boldsymbol{v}_e \rangle} \tag{83}
$$

$$
1 - 3\rho^{(t)}\beta_d^3 \langle \boldsymbol{w}^{(t)}, \boldsymbol{v}_d \rangle \geq 1 - 3\rho^{(t)}\alpha\beta_e^3 \langle \boldsymbol{w}^{(t)}, \boldsymbol{v}_e \rangle \tag{84}
$$

$$
\alpha\beta_e^3 \langle \boldsymbol{w}^{(t)}, \boldsymbol{v}_e \rangle \geq \beta_d^3 \langle \boldsymbol{w}^{(t)}, \boldsymbol{v}_d \rangle \tag{85}
$$

The last inequality holds due to $\alpha\beta_e^3 > \beta_d^3$ and $\langle \boldsymbol{w}^{(t)}, \boldsymbol{v}_e \rangle \geq \langle \boldsymbol{w}^{(t)}, \boldsymbol{v}_d \rangle$ from Theorem A.16. $\qquad \square$

From Equation 81 in the above proof, it can be seen clearly that amplifying the slow-learnable feature strength in either GD or SAM, i.e., increasing $\beta_d$, places a larger weight on the slow-learnable feature. Thus, we have the next theorem.

**Theorem A.18** (Restatement of Theorem 3.5). *Consider the training dataset $D = \{(\boldsymbol{x}_i, y_i)\}_{i=1}^N$ that follows the data distribution $\mathcal{D}(\beta_e, \beta_d, \alpha)$ using the two-layer nonlinear CNN model initialized with $\boldsymbol{W}^{(0)} \sim \mathcal{N}(0, \sigma_0^2)$. Assume that the noise is sufficiently small (ref. Lemmas B.4 and B.9) and the fast-learnable feature strength is significantly larger $\alpha^{1/3}\beta_e > \beta_d$. We have the following results for one-step upsampling, i.e. increasing $\beta_d$, from any iteration $t$ during early training*

    *1. The normalized gradient of the one-step SAM update has a larger weight on the slow-learnable feature compared to that of GD.*

    *2. Amplifying the slow-learnable feature strength puts a larger weight on the slow-learnable feature in the normalized gradients of GD and SAM.*

    *3. There exists an upsampling factor $k$ such that the normalized gradient of the one-step GD update on $\mathcal{D}(\beta_e, k\beta_d, \alpha)$ recovers the normalized gradient of the one-step SAM update on $\mathcal{D}(\beta_e, \beta_d, \alpha)$.*

*Proof.* The first result has already been proved in Lemma A.17. Now, consider increasing the slow-learnable feature strength from $\beta_d$ to $\beta_d'$. Similar to the proof of Corollary A.17, to verify that the normalized of the new normalized gradient of GD favors the slow-learnable feature, it is sufficient to show

$$\frac{(\beta_d')^3 \langle \boldsymbol{w}^{(t)}, \boldsymbol{v}_d \rangle^2}{\beta_e^3 \langle \boldsymbol{w}^{(t)}, \boldsymbol{v}_e \rangle^2} \geq \frac{\beta_d^3 \langle \boldsymbol{w}^{(t)}, \boldsymbol{v}_d \rangle^2}{\beta_e^3 \langle \boldsymbol{w}^{(t)}, \boldsymbol{v}_e \rangle^2} \tag{86}$$

which is trivial because $\beta_d' > \beta_d$. Similarly, we can verify the result for SAM. Now, let's find the new coefficient $\beta_d' = k\beta_d (k > 1)$ such that training one-step GD on $\mathcal{D}(\beta_e, \beta_d', \alpha)$ can recover the normalized gradient of the one-step SAM update on the original data distribution $\mathcal{D}(\beta_e, \beta_d, \alpha)$. Using Equation 81, we have

$$\frac{\beta_d^3 \langle \boldsymbol{w}_{\boldsymbol{\epsilon}}^{(t)}, \boldsymbol{v}_d \rangle^2}{\beta_e^3 \langle \boldsymbol{w}_{\boldsymbol{\epsilon}}^{(t)}, \boldsymbol{v}_e \rangle^2} = \frac{(\beta_d')^3 \langle \boldsymbol{w}^{(t)}, \boldsymbol{v}_d \rangle^2}{\beta_e^3 \langle \boldsymbol{w}^{(t)}, \boldsymbol{v}_e \rangle^2} \tag{87}$$

$$\frac{\langle \boldsymbol{w}_{\boldsymbol{\epsilon}}^{(t)}, \boldsymbol{v}_d \rangle}{\langle \boldsymbol{w}_{\boldsymbol{\epsilon}}^{(t)}, \boldsymbol{v}_e \rangle} = k^{3/2} \frac{\langle \boldsymbol{w}^{(t)}, \boldsymbol{v}_d \rangle}{\langle \boldsymbol{w}^{(t)}, \boldsymbol{v}_e \rangle} \tag{88}$$

$$\frac{\langle \boldsymbol{w}^{(t)}, \boldsymbol{v}_d \rangle - 3\rho^{(t)} \beta_d^3 \langle \boldsymbol{w}^{(t)}, \boldsymbol{v}_d \rangle^2}{\langle \boldsymbol{w}^{(t)}, \boldsymbol{v}_e \rangle - 3\rho^{(t)} \alpha \beta_e^3 \langle \boldsymbol{w}^{(t)}, \boldsymbol{v}_e \rangle^2} = k^{3/2} \frac{\langle \boldsymbol{w}^{(t)}, \boldsymbol{v}_d \rangle}{\langle \boldsymbol{w}^{(t)}, \boldsymbol{v}_e \rangle} \tag{89}$$

$$k^{3/2} = \frac{1 - 3\rho^{(t)} \beta_d^3 \langle \boldsymbol{w}^{(t)}, \boldsymbol{v}_d \rangle}{1 - 3\rho^{(t)} \alpha \beta_e^3 \langle \boldsymbol{w}^{(t)}, \boldsymbol{v}_e \rangle} \tag{90}$$

$$k = \left( \frac{1 - 3\rho^{(t)} \beta_d^3 \langle \boldsymbol{w}^{(t)}, \boldsymbol{v}_d \rangle}{1 - 3\rho^{(t)} \alpha \beta_e^3 \langle \boldsymbol{w}^{(t)}, \boldsymbol{v}_e \rangle} \right)^{2/3}. \tag{91}$$

Therefore, with $\beta_d' = \left( \frac{1 - 3\rho^{(t)} \beta_d^3 \langle \boldsymbol{w}^{(t)}, \boldsymbol{v}_d \rangle}{1 - 3\rho^{(t)} \alpha \beta_e^3 \langle \boldsymbol{w}^{(t)}, \boldsymbol{v}_e \rangle} \right)^{2/3} \beta_d$, the normalized gradient of the one-step GD update on $\mathcal{D}(\beta_e, \beta_d', \alpha)$ is similar to that of the one-step SAM update on $\mathcal{D}(\beta_e, \beta_d, \alpha)$. $\square$

## B   Auxiliary Lemmas

**Lemma B.1** (Claim D.20, [2]). *Considering an increasing sequence $x_t \geq 0$ defined as $x_{t+1} = x_t + \eta C_t (x_t - \rho)^2$ for some $C_t = \Theta(1), 0 \leq \rho \leq x_0$, then we have for every $A > x_0$, every $\delta \in (0, 1)$, and every $\eta \in (0, 1]$:*

$$\sum_{t \geq 0, x_t \leq A} \eta C_t \leq \frac{1 + \delta}{x_0} + \frac{O(\eta(A - \rho)^2)}{x_0^2} \frac{\log(A/x_0)}{\log(1 + \delta)} \tag{92}$$

$$\sum_{t \geq 0, x_t \leq A} \eta C_t \geq \frac{1 - \frac{(1+\delta)x_0}{A}}{x_0(1 + \delta)} - \frac{O(\eta(A - \rho)^2)}{x_0^2} \frac{\log(A/x_0)}{\log(1 + \delta)} \tag{93}$$

*Proof.* For every $g = 0, 1, \ldots$, let $T_g$ be the first iteration such that $x_t \geq (1 + \delta)^g x_0$. Let $b$ be the smallest integer such that $(1 + \delta)^b x_0 \geq A$. Suppose for notation simplicity that we replace $x_t$ with exactly $A$ whenever $x_t \geq A$. By the definition of $T_g$, we have

$$\sum_{t \in [T_g, T_{g+1})} \eta C_t [(1 + \delta)^g x_0]^2 \leq x_{T_{g+1}} - x_{T_g} \leq \delta(1 + \delta)^g x_0 + O(\eta(A - \rho)^2) \tag{94}$$

$$\sum_{t \in [T_g, T_{g+1})} \eta C_t [(1 + \delta)^{g+1} x_0]^2 \geq x_{T_{g+1}} - x_{T_g} \geq \delta(1 + \delta)^g x_0 - O(\eta(A - \rho)^2) \tag{95}$$

$$\tag{96}$$

These imply that

$$\sum_{t \in [T_g, T_{g+1})} \eta C_t \leq \frac{\delta}{(1 + \delta)^g x_0} + \frac{O(\eta(A - \rho)^2)}{x_0^2} \tag{97}$$

$$\sum_{t \in [T_g, T_{g+1})} \eta C_t \geq \frac{\delta}{(1 + \delta)^{g+2} x_0} - \frac{O(\eta(A - \rho)^2)}{x_0^2} \tag{98}$$

Recall $b$ is the smallest integer such that $(1 + \delta)^b x_0 \geq A$, so we can calculate

$$\sum_{t \geq 0, x_t \leq A} \eta C_t \leq \sum_{g=0}^{b-1} \frac{\delta}{(1 + \delta)^g x_0} + \frac{O(\eta(A - \rho)^2)}{x_0^2} b \tag{99}$$

$$= \frac{\delta}{1 - \frac{1}{1+\delta}} \frac{1}{x_0} + \frac{O(\eta(A - \rho)^2)}{x_0^2} b \tag{100}$$

$$= \frac{1 + \delta}{x_0} + \frac{O(\eta(A - \rho)^2)}{x_0^2} \frac{\log(A/x_0)}{\log(1 + \delta)} \tag{101}$$

$$\sum_{t \geq 0, x_t \leq A} \eta C_t \geq \sum_{g=0}^{b-2} \frac{\delta}{(1 + \delta)^{g+2} x_0} - \frac{O(\eta(A - \rho)^2)}{x_0^2} b \tag{102}$$

$$= \frac{\delta(1 + \delta)^{-1}(1 - \frac{1}{(1+\delta)^{(b-1)}})}{1 - \frac{1}{1+\delta}} \frac{1}{x_0} - \frac{O(\eta(A - \rho)^2)}{x_0^2} b \tag{103}$$

$$= \frac{1 - \frac{(1+\delta)x_0}{A}}{x_0(1 + \delta)} - \frac{O(\eta(A - \rho)^2)}{x_0^2} \frac{\log(A/x_0)}{\log(1 + \delta)} \tag{104}$$

Thus, the two desired inequalities are proved. $\qquad\square$

**Lemma B.2** (Lemma D.19, [2].). *Let $\{x_t, y_t\}_{t=1,\ldots}$ be two positive sequences that satisfy*

$$x_{t+1} \geq x_t + \eta \cdot C_t(x_t - \rho)^2,$$

$$y_{t+1} \leq y_t + S\eta \cdot C_t y_t^2,$$

*for some $C_t = \Theta(1)$. Suppose $x_0 \geq y_0 S \frac{1+2G}{1-3G}$ where $S \in (0, 1), G \in (0, 1/3)$ and $0 < \eta \leq \min\{\frac{G^2 x_0}{\log(A/x_0)}, \frac{G^2 y_0}{\log(1/G)}\}, 0 \leq \rho < O(x_0)$, and for all $A \in (x_0, O(1)]$, let $T_x$ be the first iteration such that $x_t \geq A$. Then, we have $y_{T_x} \leq O(G^{-1} y_0)$.*

*Proof.* Let $T_x$ be the first iteration $t$ in which $x_t \geq A$. Apply Lemma B.1 for the $x_t$ sequence with $C_t = C_t$ and threshold $A$, we have

$$\sum_{t=0}^{T_x} \eta C_t \leq \frac{1 + \delta}{x_0} + \frac{O(\eta(A - \rho)^2)}{x_0^2} \frac{\log(A/x_0)}{\log(1 + \delta)} \tag{105}$$

$$= \frac{1 + \delta}{x_0} + O\left(\frac{\eta(A - \rho)^2 \log(A/x_0)}{\delta x_0^2}\right) \tag{106}$$

$$\leq \frac{1 + \delta}{x_0} + O\left(\frac{\eta \log(A/x_0)}{\delta x_0^2}\right) \tag{107}$$

Let $T_y$ be the first iteration $t$ in which $y_t \geq A$. Apply Lemma B.1 for the $y_t$ sequence with $\eta = S\eta, C_t = C_t, \rho = 0$ and threshold $A' = G^{-1}y_0$, we have

$$\sum_{t=0}^{T_y} S\eta C_t \geq \frac{1 - \frac{(1+\delta)y_0}{A'}}{y_0(1+\delta)} - \frac{O(S\eta(A')^2)}{y_0^2} \frac{\log(A'/y_0)}{\log(1+\delta)} \tag{108}$$

$$\geq \frac{1 - O(\delta + G)}{y_0} - O\left(\frac{S\eta(A')^2 \log(1/G)}{\delta y_0^2}\right) \tag{109}$$

$$\geq \frac{1 - O(\delta + G)}{y_0} - O\left(\frac{S\eta \log(1/G)}{\delta y_0^2}\right) \tag{110}$$

Compare Equation 107 and 110. Choosing $\delta = G$ and $\eta \leq \min\{\frac{G^2 x_0}{\log(A/x_0)}, \frac{G^2 y_0}{\log(1/G)}\}$, together with $x_0 \geq y_0 S \frac{1+2G}{1-3G}$ we have $T_x \leq T_y$. $\qquad\square$

**Lemma B.3** (Lemma K.15, [32].). *Let $\{z_t\}_{t=0}^T$ be a positive sequence defined by the following recursions*

$$z_{t+1} \geq z_t + m(z_t - \rho)^2,$$
$$z_{t+1} \leq z_t + M(z_t)^2,$$

*where $z_0 > \rho \geq 0$ is the initialization and $m, M > 0$ are some constants. Let $v > z_0$, then the time $T_v$ such that $z_{T_v} \geq v$ for all $t \geq T_v$ is*

$$T_v = \frac{2z_0}{m(z_0 - \rho)^2} + \frac{4Mz_0^2}{m(z_0 - \rho)^2}\left\lceil\frac{\log(v/z_0)}{\log(2)}\right\rceil. \tag{111}$$

*Proof.* Let $n \in \mathbb{N}^\star$. Let $T_n$ be the first time that $z_t \geq 2^n z_0$. We want to find an upper bound of $T_n$. We start with the case $n = 1$. By summing the recursion, we have:

$$z_{T_1} \geq z_0 + m \sum_{t=0}^{T_1 - 1} (z_t - \rho)^2 \tag{112}$$

Because $z_t \geq z_0$, we obtain

$$T_1 \leq \frac{z_{T_1} - z_0}{m(z_0 - \rho)^2} \tag{113}$$

Now, we want to bound $z_{T_1} - z_0$. Using again the recursion and $z_{T_1 - 1} \leq 2z_0$, we have

$$z_{T_1} \leq z_{T_1 - 1} + M(z_{T_1 - 1})^2 \leq 2z_0 + 4Mz_0^2. \tag{114}$$

Combining Equation 113 and 114, we get a bound on $T_1$ as

$$T_1 \leq \frac{z_0 + 4Mz_0^2}{m(z_0 - \rho)^2} = \frac{z_0}{m(z_0 - \rho)^2} + \frac{4Mz_0^2}{m(z_0 - \rho)^2} \tag{115}$$

Now, let's find a bound for $T_n$. Starting from the recursion and using the fact that $z_t \geq 2^{n-1}z_0$ for $t \geq T_{n-1}$ we have

$$z_{T_n} \geq z_{T_{n-1}} + m \sum_{t=T_{n-1}}^{T_n - 1} (z_t - \rho)^2 \tag{116}$$

$$\geq z_{T_{n-1}} + (2^{n-1})^2 m(z_0 - \rho)^2(T_n - T_{n-1}) \tag{117}$$

On the other hand, by using $z_{T_n - 1} \leq 2^n z_0$ we upper bound $z_{T_n}$ as

$$z_{T_n} \leq z_{T_n - 1} + M(z_{T_n - 1})^2 \leq 2^n z_0 + M2^{2n}z_0^2 \tag{118}$$

Besides, we know that $z_{T_{n-1}} \geq 2^{n-1}z_0$. Therefore, we upper bound $z_{T_n} - z_{T_{n-1}}$ as

$$z_{T_n} - z_{T_{n-1}} \leq 2^{n-1}z_0 + M2^{2n}z_0^2 \tag{119}$$

Combining Equations 116 and 119 yields

$$T_n \leq T_{n-1} + \frac{2^{n-1}z_0 + M2^{2n}z_0^2}{(2^{n-1})^2 m(z_0 - \rho)^2} \tag{120}$$

$$= T_{n-1} + \frac{z_0}{2^{n-1}m(z_0 - \rho)^2} + \frac{4Mz_0^2}{m(z_0 - \rho)^2} \tag{121}$$

Summing Equation 120 for $n = 2, \dots, n$ we have

$$T_n \leq \sum_{i=1}^{n} \frac{z_0}{2^{i-1}m(z_0 - \rho)^2} + \frac{4Mnz_0^2}{m(z_0 - \rho)^2} \leq \frac{2z_0}{m(z_0 - \rho)^2} + \frac{4Mnz_0^2}{m(z_0 - \rho)^2} \tag{122}$$

Lastly, we know that $2^n z_0 \geq v$ this implies that we can set $n = \left\lceil \frac{\log(v/z_0)}{\log(2)} \right\rceil$ in Equation 122.  □

We make the following assumptions for every $t \leq T$ as the same in [32].

**Lemma B.4** (Induction hypothesis D.1, [32]). *Throughout the training process using GD for $t \leq T$, we maintain that, for every $i$ and $j \in [J]$,*

$$|\langle \boldsymbol{w}_j^{(t)}, \boldsymbol{\xi}_i \rangle| \leq \tilde{O}(\sigma_0 \sigma_p \sqrt{d}). \tag{123}$$

**Lemma B.5** (Lemma G.4, [18]). *For every $i$, we have $l_i^{(t)} = \Theta(1)g_1(t)$, where*

$$g_1(t) = sigmoid\left( \sum_{j=1}^{J} -\beta_d^3 \langle \boldsymbol{w}_j^{(t)}, \boldsymbol{v}_d \rangle^3 - \beta_e^3 \langle \boldsymbol{w}_j^{(t)}, \boldsymbol{v}_e \rangle^3 \right). \tag{124}$$

**Lemma B.6** (Lemma K.5, [32]). *Let $X \in \mathbb{R}^d$ be a Gaussian random vector, $X \sim \mathcal{N}(0, \sigma^2 I_d)$. Then with probability at least $1 - o(1)$, we have $\|X\|_2^2 = \Theta(\sigma^2 \sqrt{d})$.*

**Lemma B.7** (Lemma K.7, [32]). *Let $X$ and $Y$ be independent Gaussian random vectors on $\mathbb{R}^d$ and $X \sim \mathcal{N}(0, \sigma^2 \boldsymbol{I}_d)$, $Y \sim \mathcal{N}(0, \sigma_0^2 \boldsymbol{I}_d)$. Assume that $\sigma \sigma_0 \leq \frac{1}{d}$. Then, with probability at least $1 - \delta$, we have*

$$|\langle X, Y \rangle| \leq \sigma \sigma_0 \sqrt{2d \log \frac{2}{\delta}}$$

**Lemma B.8** (Bound on noise inner products). *Let $N = O(poly(d))$. The following hold with probability at least $1 - o(1)$:*

$$\max\left\{ |\langle \boldsymbol{w}_{j,\boldsymbol{\epsilon}}^{(0)}, \boldsymbol{\xi}_i \rangle| \right\} = \tilde{O}(\sigma \sigma_0 \sqrt{d})$$

$$\max_i \left\{ \frac{1}{n} \sum_{k=1}^{n} |\langle \boldsymbol{\xi}_k, \boldsymbol{\xi}_i \rangle| \right\} = \tilde{O}(\frac{\sigma^2 d}{N} + \sigma^2 \sqrt{d})$$

*Proof.* For the first inequality, Lemma B.7 implies that with probability at least $1 - \frac{1}{dN}$,

$$|\langle \boldsymbol{w}_{j,\boldsymbol{\epsilon}}^{(0)}, \boldsymbol{\xi}_i \rangle| \leq \sigma \sigma_0 \sqrt{2d \log(\frac{2}{dN})} = \tilde{O}(\sigma \sigma_0 \sqrt{d}) \tag{125}$$

Taking a union bound over $n = 1, \dots, N$ gives the result.

The second statement is proved similarly.  □

**Lemma B.9** (Bound on the noise component for SAM). *Assume that $\rho = o(\sigma_0)$ and $\omega(1) \leq N \leq O(poly(d))$. Throughout the training process using SAM for $t \leq T$, we maintain that, for every $i$ and $j \in [J]$,*

$$|\langle \boldsymbol{w}_{j,\boldsymbol{\epsilon}}^{(t)}, \boldsymbol{\xi}_i \rangle| \leq \tilde{O}(\sigma \sigma_0 \sqrt{d}) \tag{126}$$

*Proof.* Let $\chi_t = \max\{|\langle \boldsymbol{w}_{j,\boldsymbol{\epsilon}}^{(t)}, \boldsymbol{\xi}_i \rangle|\}$, $\alpha = \max_i\{\frac{1}{n}\sum_{k=1}^n |\langle \boldsymbol{\xi}_k, \boldsymbol{\xi}_i \rangle|\}$. Combined with $l_{k,\epsilon}^{(t)} \le 1$, the noise gradient update rule can be bounded as

$$\chi_{t+1} \le \chi_t + 3\eta\alpha\chi_t^2$$

Suppose that $a(t)$ satisfies the differential equation

$$a' = 3\alpha\eta a^2$$
$$a(0) = \chi_0$$

Observe that $a(t)$ is increasing so, by the Mean Value Theorem there exists $\tau \in (t, t+1)$ such that

$$\begin{aligned}
a(t+1) - a(t) &= a'(\tau) \\
&= 3\alpha\eta a(\tau)^2 \\
&\ge 3\alpha\eta a(t)^2
\end{aligned}$$

So an fast-learnable induction shows that $a(t) \ge \chi_t$.

Now solving for $a(t)$,

$$a(t) = \frac{1}{\frac{1}{a(0)} - 3\alpha\eta t}, \qquad t \le \frac{1}{3\alpha\eta a(0)}.$$

Using the high probability tail bounds B.8,

$$a(0) = \chi_0 = \tilde{O}(\sigma\sigma_0\sqrt{d})$$
$$\alpha = \tilde{O}(\frac{\sigma^2 d}{N} + \sigma^2\sqrt{d})$$

where $\sigma = \frac{\sigma_p}{\sqrt{d}}$. Substituting these bounds gives

$$a(t) = \frac{1}{\tilde{\Omega}(\frac{1}{\sigma\sigma_0\sqrt{d}}) - \tilde{O}(\eta t(\frac{\sigma^2 d}{N} + \sigma^2\sqrt{d}))}$$

Now Theorem A.15 and $\rho = o(\sigma_0)$ implies that $\eta T_0 = \tilde{\Theta}(\frac{1}{\sigma_0\beta_\epsilon^3})$. Combined with the assumption that $N = \Omega(1)$, the second term in the denominator is of lower order than the first term, so

$$a(T_0) = \tilde{O}(\sigma\sigma_0\sqrt{d}).$$

We conclude that

$$|\langle \boldsymbol{w}_j^{(t)}, \boldsymbol{\xi}_i \rangle| \le \tilde{O}(\sigma_0\sigma_p\sqrt{d}). \tag{127}$$

$\square$

**Lemma B.10.** *For every $i$, we have $l_i^{(t)} = \Theta(1)g_1(t)$ and $l_{i,\boldsymbol{\epsilon}}^{(t)} = \Theta(1)g_2(t)$, where*

$$g_1(t) = sigmoid\left(\sum_{j=1}^J -\beta_d^3 \langle \boldsymbol{w}_j^{(t)}, \boldsymbol{v}_d \rangle^3 - \beta_e^3 \langle \boldsymbol{w}_j^{(t)}, \boldsymbol{v}_e \rangle^3\right), \tag{128}$$

$$g_2(t) = sigmoid\left(\sum_{j=1}^J -\beta_d^3 \langle \boldsymbol{w}_{j,\boldsymbol{\epsilon}}^{(t)}, \boldsymbol{v}_d \rangle^3 - \beta_e^3 \langle \boldsymbol{w}_{j,\boldsymbol{\epsilon}}^{(t)}, \boldsymbol{v}_e \rangle^3\right). \tag{129}$$

The proof is the same as [18][Lemma G.4].

## C   Additional Experimental Settings

### C.1   Datasets and Training Details

**Datasets.** The CIFAR10 dataset [41] consists of 60,000 $32 \times 32$ color images in 10 classes, with 6000 images per class. The CIFAR100 dataset [41] is just like the CIFAR10, except it has 100 classes

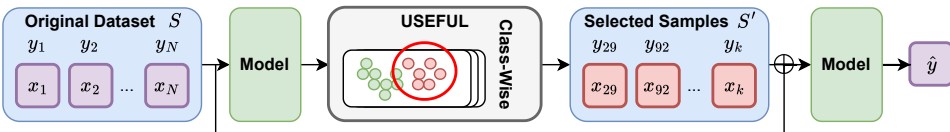

Figure 7: USEFUL first trains the model for a few epochs $t$, which in practice is around 5-10% of the total training epochs. It then clusters examples in every class into 2 groups and upsamples the cluster with higher average loss. Finally, the base model is retrained from scratch on the modified data distribution.

containing 600 images each. For both of these datasets, the training set has 50,000 images (5,000 per class for CIFAR10 and 500 per class for CIFAR100) with the test set having 10,000 images. CINIC10 [16] represents an image classification dataset consisting of 270,000 images, which is 4.5 times larger than CIFAR10. The dataset is created by merging CIFAR10 with images extracted from the ImageNet database, specifically selecting and downsampling images from the same 10 classes present in CIFAR10. Tiny-ImageNet [43] comprises 100,000 images distributed across 200 classes of ImageNet [17], with each class containing 500 images. These images have been resized to 64×64 dimensions and are in color. The dataset consists of 500 training images, 50 validation images, and 50 test images per class. The STL10 dataset [13] includes 5000 96x96 training labeled images, 500 per CIFAR10 class. The test set consists of 800 images per class, this counts up to 8,000 images in total.

**Training on different datasets.** Follow the setting from [3, 4], we trained Pre-Activation ResNet18 on all datasets except for CIFAR100 which was trained with ResNet34. We trained our models for 200 epochs with a batch size of 128 and used basic data augmentations such as random mirroring and random crop. We used SGD with the momentum parameter of 0.9 and set weight decay to 0.0005. We also fixed $\rho = 0.1$ for SAM unless further specified. For all datasets, we used a learning rate schedule where we set the initial learning rate to 0.1. The learning rate is decayed by a factor of 10 after 50% and 75% epochs, i.e., we set the learning rate to 0.01 after 100 epochs and to 0.001 after 150 epochs.

**Training with different architectures.** We used the same training procedures for Pre-Activation ResNet18, VGG19, and DenseNet121. We directly used the official Pytorch [54] implementation for VGG19 and DenseNet121. For 3-layer MLPs, we used a hidden size of 512 with a dropout of 0.1 to avoid overfitting and set $\rho = 0.01$. For ViT-S [77], we adopted a Pytorch implementation at https://github.com/lucidrains/vit-pytorch. In particular, the hidden size, the depth, the number of attention heads, and the MLP size are set to 768, 8, 8, and 2304, respectively. We adjusted the patch size to 4 to fit the resolution of CIFAR10 and set both the initial learning rate and $\rho$ to 0.01.

**Computational resources.** Each model is trained on 1 NVIDIA RTX A5000 GPU.

### C.2 Other Implementation Details

**When to separate the examples?** We selected the best-separating epoch $t$ in the set of $\{4, 5, 6, 7, 8, 10\}$ for CIFAR10 and $\{12, 14, 16, 18, 20, 22\}$ for CIFAR100. Particularly, we separated examples of CIFAR10 at around epoch 8 while that of CIFAR100 is near epoch 20. Near this point, the gain in training error diminishes significantly as shown in Figure 15a, which shows a sign that the model successfully learns fast-learnable features. In addition, we reported the results for different separating epochs in Appendix D.8. In Figure 4, the best separating epochs for STL10, CINIC10, and Tiny-ImageNet are 11, 4, and 10, respectively. The separating epoch for Waterbirds is 5.

**Forgetting score.** To compute forgetting scores of training examples in each dataset, we collected the same statistics as in [69] but computed at the end of each epoch. The reason is to make the statistics consistent between two versions of the same slow-learnable example which is repeated in the upsampled dataset.

**Hessian spectra.** We approximated the density of the Hessian spectrum using the Lanczos algorithm [24, 53]. The Hessian matrix is approximated by 1000 examples (100 per class of CIFAR10). Then we extract the top eigenvalues to calculate the maximum Hessian eigenvalue ($\lambda_{\max}$) and the bulk of spectra ($\lambda_{\max}/\lambda_5$) [30].

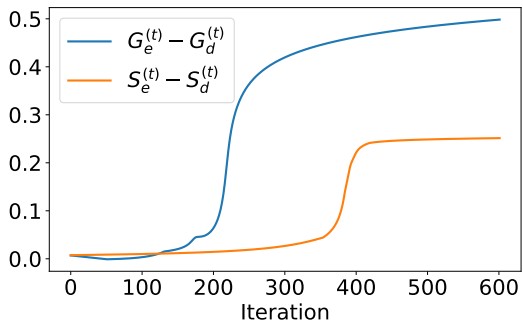

Figure 8: The gap between contribution of fast-learnable and slow-learnable features towards the model output in SAM and GD. The toy datasets is generated from the distribution in Definition 3.1 with $\beta_d = \beta_e = \alpha = 1$.

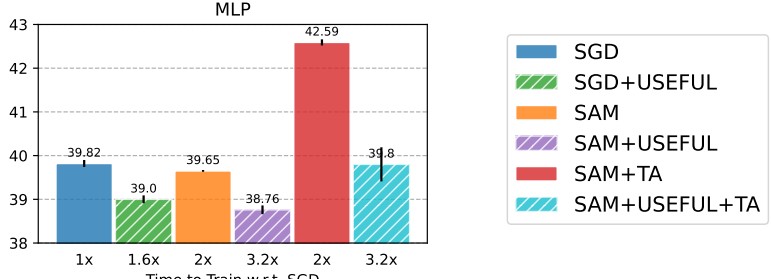

Figure 9: **Test classification errors of 3-layer MLP on CIFAR10.** The number below each bar indicates the estimated cost to train the model and the tick on top shows the standard deviation over three runs. USEFUL improves the performance of SGD and SAM when training with 3-layer MLP.

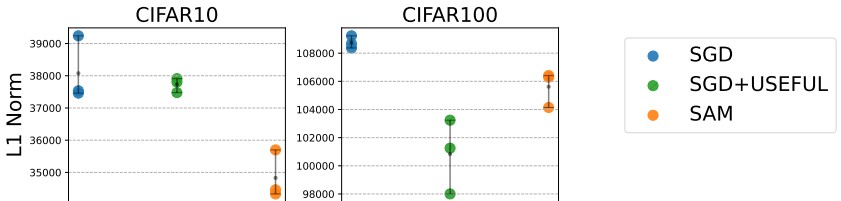

Figure 10: **L1 norm of ResNet18 trained on CIFAR10 and ResNet34 trained on CIFAR100.** Lower L1 norm indicates a sparser solution and stronger implicit regularization properties [3, Section 4.2]. SAM has a lower L1 norm than SGD, and USEFUL further reduces the L1 norm of SGD and SAM.

## D    Additional Results

### D.1    An Extreme Case of Toy Datasets

We consider an extreme setting when two features have identical strength and no missing fast-learnable features, i.e., $\beta_e = \beta_d = \alpha = 1$, the gap between LHS and RHS in Equation 5 is not small as shown in the figure 8. The gap is consistently around 0.2-0.3 from epoch 250 onwards.

### D.2    USEFUL is Useful for MLP

Figure 9 shows that for 3-layer MLP, USEFUL successfully reduces the test error of both SGD and SAM by nearly 1%. Additionally, SGD+USEFUL yields better performance than SAM alone.

Table 1: **Sharpness of solution at convergence.** We train ResNet18 on CIFAR10 and measure the maximum Hessian eigenvalue $\lambda_{\max}$ and the bulk spectra measured as $\lambda_{\max}/\lambda_5$.

| METRIC | SGD | SGD+USEFUL | SAM |
|---|---|---|---|
| $\lambda_{\max}$ | 53.8 | 41.8 | 12.4 |
| $\lambda_{\max}/\lambda_5$ | 3.8 | 1.5 | 2.4 |

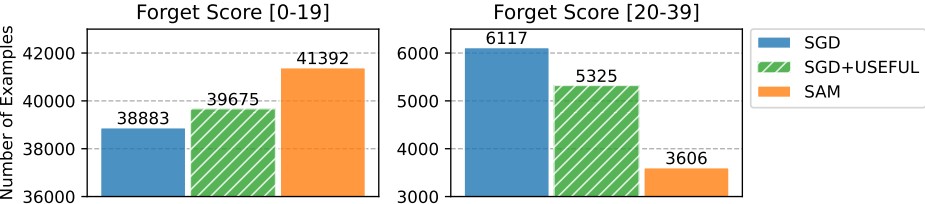

Figure 11: **Forgetting scores for training ResNet18 on CIFAR10.** Forgetting scores measure the learning speed of examples in training data. USEFUL approaches the training dynamics of SAM, with more examples being forgotten infrequently and fewer examples being forgotten frequently.

Table 2: Average scores for two clusters on CIFAR10.

| METRIC | FAST-LEARNABLE | SLOW-LEARNABLE |
|---|---|---|
| FORGETTING SCORE | $3.8 \pm 6.1$ | $14.7 \pm 9.0$ |
| FIRST LEARNED ITERATION | $0.9 \pm 1.2$ | $3.7 \pm 8.0$ |
| ITERATION LEARNED | $45.6 \pm 50.0$ | $105.8 \pm 41.0$ |

Table 3: Average scores for two clusters on CIFAR100.

| METRIC | FAST-LEARNABLE | SLOW-LEARNABLE |
|---|---|---|
| FORGETTING SCORE | $10.2 \pm 8.6$ | $16.8 \pm 7.4$ |
| FIRST LEARNED ITERATION | $4.7 \pm 6.8$ | $9.8 \pm 13.7$ |
| ITERATION LEARNED | $86.6 \pm 46.1$ | $115.7 \pm 31.0$ |

## D.3 SAM & USEFUL Reduce Forgetting Scores

We used forgetting scores [69] to partition examples in CIFAR10 into different groups. Forgetting scores count the number of times an example is misclassified after being correctly classified during training and is an indicator of the learning-speed of examples. Figure 11 illustrates that SGD+USEFUL and SAM have fewer examples with high forgetting scores than SGD does. This aligns with our theoretical analysis in Theorem 3.4 and results on the toy datasets. By upsampling slow-learnable examples in the dataset, they contribute more to learning and hence SGD+USEFUL learns slow-learnable features faster than SGD.

Furthermore, Tables 2 and 3 illustrate the average forgetting score, first learned iteration (i.e., at this epoch, the model predicts correctly for the first time) and iteration learned (i.e., after this epoch, the prediction is always correct) of examples in fast-learnable and slow-learnable clusters. Iteration learned is highly correlated with prediction depth [5], which is another notion of data difficulty. It can be seen clearly that examples in fast-learnable clusters have a lower difficulty score for every metric, indicating that USEFUL successfully identifies fast-learnable examples early in training.

Table 4: Test classification errors for training SAM and ASAM on the original CIFAR10 and modified datasets by USEFUL. Results are averaged over 3 seeds.

|              | SAM           | ASAM          |
| ------------ | ------------- | ------------- |
| +            | $4.23 \pm 0.08$ | $4.33 \pm 0.19$ |
| + USEFUL     | **$4.04 \pm 0.06$** | **$4.09 \pm 0.10$** |
| + TA         | $4.06 \pm 0.08$ | $3.93 \pm 0.11$ |
| + TA + USEFUL | **$3.49 \pm 0.09$** | **$3.46 \pm 0.01$** |

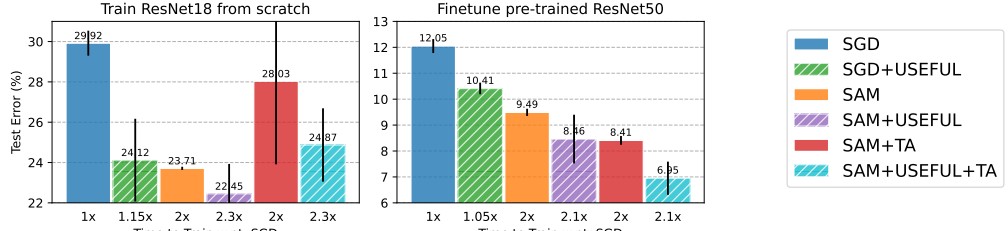

Figure 12: **Comparing test classification errors on Waterbirds.** The number below each bar indicates the approximated cost to train the model and the tick on top shows the standard deviation over three runs. USEFUL boosts the performance of SGD and SAM on the balanced test set, showing its generalization to the OOD setting. In addition, the success of USEFUL in fine-tuning reveals its new application to the transfer learning setting.

Table 5: Comparison between USEFUL and upweighting loss regarding test classification errors. Upweighting loss doubled the loss for all examples in the slow-learnable clusters found by USEFUL, which is different from dynamically upweighting examples during the training [78]. Results are averaged over 3 seeds.

| METHOD           | SGD           |               | SAM           |               |
| ---------------- | ------------- | ------------- | ------------- | ------------- |
|                  | CIFAR10       | CIFAR100      | CIFAR10       | CIFAR100      |
| UPWEIGHTING LOSS | $5.33 \pm 0.09$ | $26.70 \pm 3.25$ | $4.28 \pm 0.02$ | $21.75 \pm 0.25$ |
| USEFUL           | **$4.79 \pm 0.05$** | **$22.58 \pm 0.08$** | **$4.04 \pm 0.06$** | **$20.40 \pm 0.36$** |

## D.4 USEFUL generalizes to other SAM variants

In this experiment, we show that SAM can also generalize to other variants of SAM. We chose ASAM, which is proposed by Kwon et al. to address the sensitivity of parameter re-scaling [19]. Following the recommended settings in ASAM, we trained it with a perturbation radius $\rho = 1.0$, which is 10 times that of SAM. Other settings are identical to the standard settings in Appendix C. Table 4 demonstrates the results for training ResNet18 on CIFAR10. Both SAM and ASAM can be combined with USEFULto improve the test classification error. When using TA, ASAM shows a slightly better performance than SAM.

## D.5 USEFUL shows promising results for the OOD settings

While our main contribution is providing a novel and effective method to improve the in-distribution generalization performance, we conducted new experiments confirming the benefits of our method to improving out-of-distribution (OOD) generalization performance. As a few very recent works [9, 72] showed the benefit of SAM in improving OOD performance, it is expected that USEFUL also extend to this setting.

**Spurious correlation (Waterbirds).** As can be seen in Figure 12, both SAM and USEFUL effectively improve the performance on the balanced test set though the model is trained on the spurious training

Table 6: Test error on long-tailed CIFAR10. BALANCING means that we upsampled small classes to make the classes balanced. Results are averaged over 3 seeds.

| RATIO | SGD | SGD+USEFUL | SAM | SAM+USEFUL |
|---|---|---|---|---|
| 1:10 | $10.01 \pm 0.21$ | $9.53 \pm 0.13$ | $8.85 \pm 0.08$ | $8.22 \pm 0.04$ |
| BALANCING | $9.77 \pm 0.17$ | $9.25 \pm 0.11$ | $8.31 \pm 0.11$ | $7.93 \pm 0.02$ |

Table 7: **Test classification errors of SGD** for different partition methods. Results are averaged over 3 seeds.

| PARTITION METHOD | CIFAR10 | CIFAR100 |
|---|---|---|
| QUANTILE | $5.27 \pm 0.10$ | $23.49 \pm 0.82$ |
| MISCLASSIFICATION | $4.98 \pm 0.17$ | $23.86 \pm 0.70$ |
| USEFUL | $\mathbf{4.79 \pm 0.05}$ | $\mathbf{22.58 \pm 0.08}$ |

set. When training ResNet18 with SGD from scratch, USEFUL decreases the classification errors by 5.8%. In addition, it can be successfully applied to the pre-trained ResNet50 (on ImageNet), opening up a promising application to the transfer learning setting.

**Long-tail distribution (Long-tail CIFAR10).** We conducted new experiments on long-tail CIFAR10 [15] with an imbalance ratio of 10. Table 6 shows that USEFUL can also improve the performance of SGD and SAM, by reducing the simplicity bias on the long-tail data. Figure 13 visualizes the distribution of classes before and after upsampling by USEFUL. Interestingly, we see that USEFUL upsamples more examples from some of the larger classes and still improves the accuracy on the balanced test set. This improvement is attributed to the more uniform speed of feature learning, and not balancing the training data distribution. Notably, USEFUL outperforms class balancing to address long tail distribution. Besides, USEFUL can be stacked with methods to address long-tail data to further improve performance, as we confirmed in the last row.

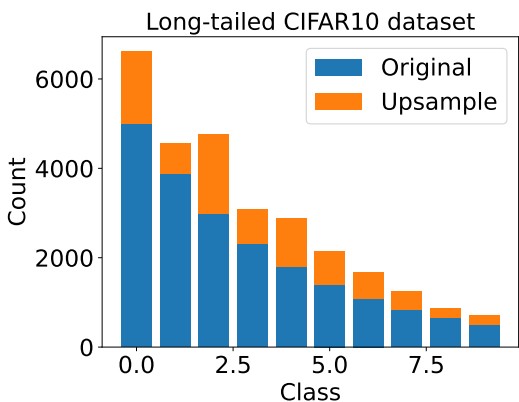

Figure 13: Class count before and after upsampling by USEFUL on long-tail CIFAR10 dataset.

## D.6  USEFUL is also effective for noisy label data

Our method and analysis consider a clean dataset. But, as we confirmed in Table 8, USEFUL can easily stack with MixUp [79], a robust method for learning against noisy labels, to reduce the simplicity bias and improve their performance. Applying USEFUL on top of MixUp to CIFAR10 with 10-20% random label flip successfully boosts the performance of both SGD and SAM when training on data with corrupt labels.

Table 8: Test error on label noise CIFAR10 (all methods are with MixUp). Results are averaged over 3 seeds.

| Rate | SGD | SGD+USEFUL | SAM | SAM+USEFUL |
|------|-----|------------|-----|------------|
| 10% | $7.20 \pm 0.17$ | $6.64 \pm 0.10$ | $5.15 \pm 0.05$ | $4.75 \pm 0.09$ |
| 20% | $9.26 \pm 0.23$ | $8.88 \pm 0.07$ | $6.08 \pm 0.06$ | $5.82 \pm 0.05$ |

Table 9: Test errors of different simplicity bias reduction methods on CIFAR10. Results are averaged over 3 seeds.

| SGD | EIIL | JTT | SGD+USEFUL |
|-----|------|-----|------------|
| $5.07 \pm 0.04$ | $5.04 \pm 0.04$ | $4.89 \pm 0.03$ | $\mathbf{4.79 \pm 0.05}$ |

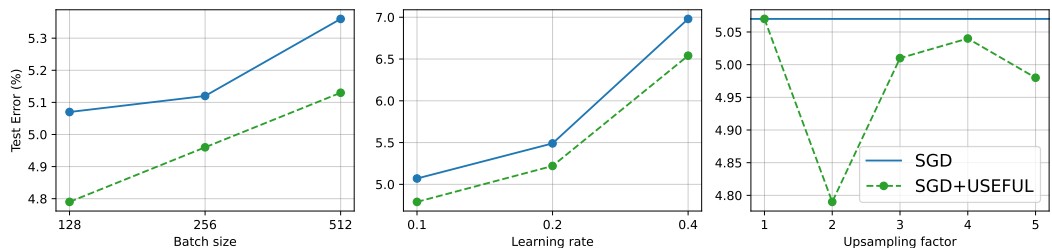

Figure 14: **Ablation studies of training ResNet18 on CIFAR10.** In each experiment, we used the standard training settings while (left) varying training batch size or (middle) varying learning rate, or (right) varying upsampling factor.

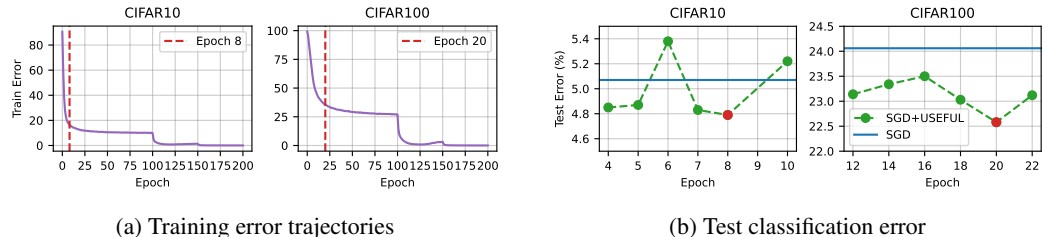

(a) Training error trajectories          (b) Test classification error

Figure 15: **Separating epoch analysis.** (left) Red lines indicate our optimal choice of $t$ to separate examples at and restart training. The early epoch that can best separate the examples is when the change in training error starts to shrink. (right) Red points indicate our optimal choice of $t$.

## D.7 Comparison with simplicity bias mitigation methods

While previous works show that reducing simplicity bias benefits the OOD settings, we show that reducing the simplicity bias also benefits the ID settings. To confirm our hypothesis on our simplicity bias mitigation baselines, we applied EIIL [14] and JTT [48] to train ResNet18 on CIFAR10. The choice of the baselines is because they have publicly available code and fewer hyperparameters to tune in our limited rebuttal time. For EIIL, we tuned lr $\in \{$1e-1, 1e-2, 1e-3, 5e-4, 1e-4$\}$, number of epochs $\in \{$1, 2, 4, 8$\}$ for training the reference model, and the weight decay $\in \{$1e-3, 5e-4, 1e-4$\}$ for training GroupDRO. For JTT, we tuned the separating epoch $\in \{$4, 5, 6, 7, 8, 10$\}$ and upsampling factor $\in \{$2, 3, 4, 5$\}$, while lr and weight decay follow the standard training of ResNet18 on CIFAR10. Table 9 shows that all methods successfully reduce the simplicity bias, yielding an improvement over SGD. While EIIL requires tuning 3 hyperparameters with a total of 60 combinations, USEFUL only requires one hyperparameter, which is the separating epoch within a small range (around the time when the slope of the training loss curve diminishes).

### D.8 Ablation studies

**USEFUL vs Upweighting loss.** We compare USEFUL with upweighting loss of examples in the slow-learnable clusters. As can be seen in Table 5, when coupling with either SGD or SAM, USEFUL clearly outperforms upweighting loss on both CIFAR datasets. It is worth mentioning that upweighting loss is different from iteratively importance sampling methods such as GRW [78], which dynamically upweights examples during the training by factors that depend on the loss value. In addition, GRW is dedicated to the distribution shift setting while our paper considers the in-distribution setting.

**Data selection method.** In this experiment, we compare clustering with other methods for partitioning data. The first baseline is to upsample misclassified examples (MISCLASSIFICATION) while the second baseline is to upsample all examples whose training errors are larger than the median value (QUANTILE). All the three methods are performed at the same epoch $t$. Table 7 shows that USEFUL selects a better set of upsampling examples, leading to the best model performance.

**Training batch size.** Figure 14 left shows the gap between USEFULand SGD when changing the batch size. Our method consistently improves the performance, proving its effectiveness is not simply captured by the gradient variance caused by the training batch size.

**Learning rate.** The small learning rate is a nuance in our theoretical results to guarantee that fast-learnable and slow-learnable features are separable in early training. In general, a small learning rate is required for most theoretical results on gradient descent and its convergence and is a standard theoretical assumption [12, 60, 62, 73]. In practice, both for separating fast-learnable vs slow-learnable examples and for training on the upsampled data, we used the standard learning rate that results in the best generalization performance for both SGD and SAM following prior work [3, 42, 81]. While the theoretical requirement for the learning rate is always smaller than the one that is used in practice, empirically a larger learning rate does not yield better generalization for the problems we considered in contrast to other settings [46, 58]. As shown in Figure 14 middle, increasing the learning rate has an adverse effect on the model performance. Indeed, for a fair comparison, the algorithms should be trained with hyperparameters that empirically yield the best performance; otherwise, the conclusions are not valid. Nevertheless, USEFUL always improves the test error across different learning rates.

**Upsampling factor.** We empirically found the upsampling factor of 2 to consistently work well across different datasets and architectures. Using larger upsampling factors results in a too-large discrepancy between the training and test distribution and does not work better, as is expected and discussed in Section 1. As illustrated in Figure 14 right, while all factors from 2 to 5 bring performance improvement, the upsampling factor of 2 yields the best performance, as it reduces the simplicity bias with minimum change to the rest of the data distribution.

**Separating epoch.** Fig 15a shows that the early epoch that can best separate the examples is when the change in training error starts to shrink. At this time, examples that are learned can be separated from the rest by clustering model outputs as analyzed in our theoretical results in Section 3.3. Figure 15b demonstrates the performance of USEFUL when separating examples at different epochs in early training. Too early or too late epochs do not cluster examples well, i.e., some examples with fast-learnable features fall into the slow-learnable clusters and vice versa. This ablation study shows that upsampling correct examples and with enough amount is important for our method to achieve its best. Note that there is no universal separating epoch. This is reasonable because each data has a different data distribution, i.e. slow-learnable and fast-learnable features, thus, a different theoretical time $T$ in Theorems 3.2 and 3.3.

