# OpenReview forum: "Changing the Training Data Distribution to Reduce Simplicity Bias Improves In-distribution Generalization"
_NeurIPS.cc/2024/Conference — NeurIPS 2024 poster_

### Official Review · Reviewer_F4yx · 2024-07-04

**Soundness:** 3
**Presentation:** 3
**Contribution:** 3
**Rating:** 6
**Confidence:** 2

**Summary:**

It is known that usually deep neural networks will learn “easy examples" that contain fast-learnable features first while learning more complex examples in a second time. The authors argue that mitigating such simplicity bias is the reason method like SAM are outperforming SGD. Based on such analysis, the authors introduce their methods coined as USEFUL that consists in two setups: 1) Identifying the examples with fast-learnable features using a clustering method based on layer output similarity 2) Upsampling by a constant factor the remaining examples with slow-learning features. By doing so, the authors can significantly increase model performances and training time on different classification tasks using different optimizers. They assess their methods across a wide range of dataset and different hyper-parameters and outperform random clustering baseline.

**Strengths:**

This paper is well motivated and written. The method seems to be sounded and I really appreciate that the authors assess their method using different hyper-parameters such as optimizer, batch size, datasets, upsampling factor, architectures, and data augmentation. It is also great that they ran a baseline with random clustering.

**Weaknesses:**

It is not clear when and why one should choose the last output activation vector to define the clustering instead of intermediate activation vector. It is also not clear at which epoch one should decide to do the clustering since for a dataset like CIFAR10 the optimal performances are achieved at epoch 8 while for CIFAR100 it is epoch 20. So, finding the correct hyper-parameters for the clustering might be costly and thus impact how fast convergence can really be (if we consider this needed additional ablation on clustering epoch). In addition, the authors mention that they are using an upscaling factor of 2, but I am wondering how robust this is when using long-tail distribution. For example, I am not sure that on something like ImageNet-LT or Inaturalist, we will get the best performances by using a constant factor.  I would also be a bit more cautious about some of the claims made in the papers. For example, the authors claim that their method is generalizing to OOD tasks while providing experiments on only the WaterBird dataset.  So, it would be better to write about promising preliminary results than claiming generalization on OOD.

**Questions:**

1) Do you think your method will also generalize on Long-Tail dataset while keeping an upscaling factor constant?

2)) Any ideas or heuristics about how to find the optimal epoch/layer to perform the clustering without running an expensive ablation?

**Limitations:**

The authors did not really discuss any limitations (outside the fact that their theoretical result does not extend to CNN) or societal impact. I think that one limitation that could have been highlighted is the smaller scale of the experiment and the focus on classification tasks. Another limitation is the lack of results on OOD or long-tail benchmarks which would seem to be well suited for this type of work.

---

> ### Author Rebuttal · Authors · 2024-08-07
>
> We thank the reviewer for the valuable feedback, and acknowledging our well-motivated work and our comprehensive experiments and ablations.
>
> 1. When and why one should choose the last output activation vector to define the clustering instead of intermediate activation vector?
> - Our theorems 3.2 and 3.3 show that examples with at least one fast-learnable feature are learned in early training iterations, and this is well reflected in the model's final output (normalized logit vector). Intuitively, regardless of the layer in which different fast-learnable features are learned, the model’s output for examples containing any fast-learnable feature become similar to its one-hot encoded label. Hence, examples with at least one fast-learnable feature can be well separated based on the model’s final output. While one may be able to separate examples with the same fast-learnable feature by clustering model’s activation at different layers, this is not necessary for our purpose, which is reducing the simplicity bias of early training. In our experiments, we used the model’s final output (output of the softmax), without further tuning the layer number.
> 2. How to find the separating epoch?
> - In Fig 14a, we showed that the epoch that can best separate the examples is when the reduction in training error starts to diminish. Intuitively, the first part of the plot with a high negative slope is the time that fast-learnable features are learned due to the simplicity bias. When the training loss curve diminishes the examples with fast learnable features can be best separated from the remaining examples. Figure 14b shows our ablation and confirms that separating examples at multiple epochs around this time (when the slope of the training loss curve diminishes) and upsampling the remaining examples outperform SGD. This observation can help find the separating epoch relatively quickly.
> 3. How robust is the scaling factor of 2 on long-tail distribution?
> - The reason for the scaling factor of 2 is to reduce the simplicity bias of early training without further modifying the training data distribution. We conducted new experiments on long tail CIFAR10 with an imbalanced ratio of 10. Our results show that our method with a scaling factor of 2 can indeed improve the generalization performance. Notably, *USEFUL outperforms class balancing to address long tail distribution*.  As can be seen in Fig 1 of the [PDF](https://openreview.net/forum?id=yySpldUsU2&noteId=nTUcV9AnO0), interestingly our method may upsample more examples from some of the larger classes (not smaller classes). This confirms that, instead of balancing the training data distribution, it benefits the performance by reducing the simplicity bias and learning features of a class at a more uniform speed. Empirically, as we confirmed in Fig 13 and our new experiments on long-tail data, a scaling factor of 2 works best to reduce the simplicity bias. We note that our method can be stacked with existing methods for long-tail data to balance the training data distribution and further improve the performance, as we confirmed in our new experiments.
>
>   **Table 3**: Test error (avg of 3 runs) on long-tailed CIFAR10.
>
>   | Ratio | SGD | SGD + USEFUL | SAM | SAM + USEFUL |
>   | --- | --- | --- | --- | --- |
>   | 1:10 | 10.01 ± 0.21 | 9.53 ± 0.13 | 8.85 ± 0.08 | 8.22 ± 0.04 |
>   | Balancing | 9.77 ± 0.17 | 9.25 ± 0.11 | 8.31 ± 0.11 | 7.93 ± 0.02 |
>
> 4. The authors claim that their method is generalizing to OOD tasks while providing experiments on only the WaterBird dataset.
> - Prior works have shown the benefits of reduced simplicity bias to improving the OOD performance [Evading the Simplicity Bias, CVPR’22, Overcoming Simplicity Bias, ICML’23, Identifying Spurious Biases Early, AISTATS’24]. Our method reduces the simplicity bias and hence is expected to benefit the OOD performance too. We confirmed this on one of the standard OOD benchmark datasets to not distract the reader from the main contribution of the paper, i.e. improved ID performance. We thank the reviewer and will revise our language as suggested.

---

> ### Author Response · Authors · 2024-08-11
> **Your feedback is greatly appreciated**
>
> We hope our rebuttal has effectively addressed your concerns. As the discussion phase is nearing its conclusion, we are wondering if there's anything further we can clarify or elaborate on. Your feedback is greatly appreciated.

---

> ### Comment · Reviewer_F4yx · 2024-08-12
>
> I would like to thank the authors for addressing my concerns. I really appreciate the new long tail experiment and the fact that the authors will revise some of the paper's language. I am maintaining my score.

---

> > ### Author Response · Authors · 2024-08-12
> > **Response to Reviewer**
> >
> > Thank you for reading our rebuttal and keeping a positive view of our paper. We are glad that our rebuttal, especially the new long-tail experiment, has addressed your concerns. We assure that we will modify our language in the revised version.

---

### Official Review · Reviewer_EFb1 · 2024-07-07

**Soundness:** 3
**Presentation:** 3
**Contribution:** 2
**Rating:** 6
**Confidence:** 3

**Summary:**

This work aims to modify the training data distribution to improve in-distribution generalization. First, the authors theoretically analyse a 2-layer CNN and compare the feature learning dynamics (fast learnable and slow-learnable features) of Gradient Descent (GD) and Sharpness-Aware Minimization (SAM). It is then shown that SAM mitigates simplicity bias compared to GD. The authors then propose USEFUL (UpSample Early For Uniform Learning), a method that upsamples the examples in the training set that contains slow-learnable features.  USEFUL first clusters the examples with similar outputs early in the training and then upsamples the slow-learnable clusters. The main idea behind USEFUL is to learn features at a uniform speed (similar to SAM) by changing the training data distribution. USEFUL can be trained with SGD, SAM and SAM + Trivial Augment. Results on CIFAR-10, CIFAR-100, STL10, TinyImageNet indicate that USEFUL is across datasets and architectures. Additonal ablation and analysis show that USEFUL learns similar properties to SAM (for e.g less sharp solutions).

**Strengths:**

1. Originality: The question posed by the authors “Can we change the training data distribution such that the model trained on it has similar properties to SAM?” is interesting and novel. The proposed method is also well-motivated.
2. Results: The authors perform a comprehensive set of ablations and analysis on the proposed method USEFUL. Section 5.4 that shows that USEFUL’s solution has similar properties to SAM, which answers the question raised in the motivation of the paper. I also particularly like the ablations with upweighting loss and data selection method in Appendix D.6.
3. Overall, the paper is fairly well written. One minor point to address here is that the paper covers multiple concepts like SAM, simplicity bias, flat minima and uniform feature learning. It would be good to explain the relationship between these more clearly.

**Weaknesses:**

1. The authors explicitly mention that their focus in this paper is only on “in-distribution generalization”. I am a bit confused by this given the motivation of simplicity bias and learning features uniformly. To elaborate more on this point,
    - Springer et al [1] also show that SAM implicitly balances the quality of diverse features (similar to the observations made in Section 3 of this paper. The experimental results in [1] is focused more on datasets with multiple predictive features like CelebA, CIFAR-MNIST.
    - Past work on simplicity bias and shortcut learning [2, 3, 4, 5] has focused on similar datasets like CelebA, Waterbirds, CIFAR-MNIST, Colored-MNIST to name a few.
    - While the authors have shown encouraging results on Waterbirds dataset in Appendix D5, it would be good to show the complete results on various groups and on other datasets as well.
2. Connection to [1]. Springer et al [1] made a very similar observation as to Section 3 in this paper. It would be great if the authors can clarify the differences with the observations in [1] and this work. Particularly, [1] also shows that SAM mitigates simplicity bias and that  SAM learns higher quality representations of hard-to-learn features. The authors briefly discuss this in Related Works section but a more detailed answer would be helpful.
3. I just wanted to understand the practical usefulness of the proposed method. This method has one additional hyperparameter i.e the separating epoch. The authors have reported the best separating epoch for all the datasets which is epoch 8 for CIFAR-10 and epoch 20 for CIFAR-100 (Appendix C.2). How is this hyperparameter chosen? Is there a separating epoch number that works across various datasets? This is especially relevant given that that the average gain on most of the datasets with USEFUL is less than 1% with additional cost for training.

 [1] Springer, Jacob Mitchell, Vaishnavh Nagarajan, and Aditi Raghunathan. "Sharpness-Aware Minimization Enhances Feature Quality via Balanced Learning." The Twelfth International Conference on Learning Representations.

[2] Shah, Harshay, et al. "The pitfalls of simplicity bias in neural networks." Advances in Neural Information Processing Systems 33 (2020): 9573-9585.

 [3] Geirhos, Robert, et al. "Shortcut learning in deep neural networks." Nature Machine Intelligence 2.11 (2020): 665-673.

 [4] Kirichenko, Polina, Pavel Izmailov, and Andrew Gordon Wilson. "Last layer re-training is sufficient for robustness to spurious correlations." arXiv preprint arXiv:2204.02937 (2022).

 [5] Teney, Damien, et al. "Evading the simplicity bias: Training a diverse set of models discovers solutions with superior ood generalization." Proceedings of the IEEE/CVF conference on computer vision and pattern recognition. 2022.

**Questions:**

1. I did not find the separating epoch used for most of the datasets (except CIFAR-10 and CIFAR-100). Could you please point me to that?
2. What is the takeaway from Figure 1? Please mention the observations regarding the Figure involving the fast and slow learnable features. What kind of examples are usually clustered in fast learnable cluster vs slow learnable cluster,

Please refer to the Weakness section for remaining questions.

**Limitations:**

Yes, the authors have addressed limitations.

---

> ### Author Rebuttal · Authors · 2024-08-07
>
> We thank the reviewer for their valuable feedback and recognizing the originality and novelty of our work, and our comprehensive experiments.
>
> 1. Our work (ID) vs [1-5] (OOD).
> - As the reviewers correctly mentioned and we discussed in our [general comment](https://openreview.net/forum?id=yySpldUsU2&noteId=nTUcV9AnO0), our work studies ID, while prior work including [1-5] studied OOD with spurious features (CelebA, Waterbirds, CMNIST and CIFAR-MNIST are all benchmark datasets with spurious features in training data but not on the test data. Thus spurious features are not predictive on test). In the OOD setting, simple spurious (non-predictive) features are learned from the training data *instead of the predictive features*, due to simplicity bias. Training with SAM [1] or methods for alleviating the simplicity bias [2-5] suppresses learning the spurious features and allows learning *more/other* predictive features. In the ID setting (CIFAR10-100, TinyImageNet), there is no spurious feature and all features are predictive on the test set. Hence, it is not clear why reducing the simplicity bias should help. We proved that reducing the simplicity bias allows learning the same set of predictive features (not new features) at a more uniform speed. This benefits the ID performance. To our knowledge, **our study is the first to show the benefits of reduced simplicity bias to ID, hence is a major contribution**. Our results on Waterbirds shows that our method also benefits OOD, but is not the main contribution of our work (as we mentioned in lines 338-339).
> 2. How is the separating epoch chosen?
> - In Fig 14a, we showed that the epoch that can best separate the examples is when the reduction in training error starts to diminish. Intuitively, the first part of the plot with a high negative slope is the time that fast-learnable features are learned due to the simplicity bias. When the training loss curve diminishes the examples with fast learnable features can be best separated from the remaining examples. Figure 14b shows that separating examples at multiple epochs around this time (when the slope of the training loss curve diminishes) and upsampling the remaining examples outperform SGD. We report the best-performing separating epoch (red points in the figures) in our experiments.
> - Is there a separating epoch number that works across various datasets?
>     - There is no universal separating epoch. This is reasonable because each data has a different data distribution, i.e. slow- and fast-learnable features, thus, a different theoretical time T in Theorems 3.2 and 3.3.
> - The average gain on most of the datasets with USEFUL is less than 1% with additional cost for training.
>     - Fig 4, 5 show that SAM’s improvement over SGD is also around 1%. While SAM increases the training time by 2x, and requires tuning the inner step size $\rho$ for different model architectures and datasets, it attracted a lot of attention and is considered a very important contribution as it improves the SOTA performance. Our method is much cheaper (~1.3x cost) than SAM (2x cost) and can be easily stacked with SAM and TA to further improve the SOTA performance. Thus, we believe our contribution is novel and important and can lead to a line of follow up work.
>
> **Questions**:
>
> 1. **Separating epoch** used for all datasets
> - In Figure 4, the best separating epochs for STL10, CINIC10, and Tiny-ImageNet are 11, 4, and 10, respectively. The separating epoch for Waterbirds is 5. We will add these details into our revision.
> 2. What is the takeaway from Figure 1? What kind of examples are usually clustered in fast learnable cluster vs slow learnable cluster?
> - The bottom row in Fig 1 shows examples with fast learnable features that are learned in the first few epochs and are found by our method. We see that these examples are not ambiguous and are clear representatives of their corresponding class, hence are very *easy* to visually classify (the entire object is in the image and the area associated with the background is small). Such examples are learned early in training due to the simplicity bias. In contrast, the top row shows that examples without any fast learnable feature are *harder,* to identify visually and look more ambiguous (part of the object is in the image or the object is smaller and the area associated with the background is larger). These examples still contain features that are predictive on the test set, but are learned later during the training (require more iterations to be learned). Our method enables learning both types of examples containing fast-learnable and easy-learnable features at a more uniform speed, which we showed benefits the ID performance.

---

> > ### Comment · Reviewer_EFb1 · 2024-08-09
> > **Response to Rebuttal**
> >
> > I thank the authors for the detailed rebuttal. The authors have clearly answered most of the questions.
> >
> > > reducing the simplicity bias benefits in-distribution (ID).
> >
> > This is an interesting result and a bold claim.
> > Does this mean that past methods that focus on mitigating simplicity bias (and show results in the OOD scenarios) also improve the performance in IID setup by similar margins? The additional simplicity bias (JTT and EIIL) baselines discussed in the rebuttal pdf seem to indicate so. I would love to hear the authors' comments on this. In the future, the authors can consider running experiments with a couple of other methods that mitigate simplicity to verify the generality of the claim.
> >
> > Most of my concerns have been resolved and thus, I have increased my score to 6.

---

> ### Author Response · Authors · 2024-08-09
> **Response to Reviewer**
>
> Thank you for reading our rebuttal, and we’re glad that you found our findings interesting and striking. We also believe that this is an important result and we are quite excited about it.
>
> As shown by our theory, we expect prior methods for reducing the simplicity bias to also benefit ID, and our preliminary results with JTT and EIIL showed their promise. Such methods often have several hyperparameters that need to be carefully tuned via a grid search, as we listed for JTT and EIIL in our rebuttal. Our method is directly motivated by our theory and requires tuning only one hyperparameter within a small range (guided by the training loss curve). Hence, we expect it to be more effective in the ID setting. Nevertheless, we agree that adding prior simplicity bias methods would be a nice addition to our work and further support the finding and generality of our result. We thank the reviewer for their valuable suggestion, and we will add experiments with more simplicity bias methods to our revised version.

---

### Official Review · Reviewer_eKCz · 2024-07-14

**Soundness:** 3
**Presentation:** 3
**Contribution:** 2
**Rating:** 4
**Confidence:** 4

**Summary:**

- Proves for a 2-layer CNN with fixed second layer weighsts trained on a toy dataset,  SAM learns slow-learnable and fast-learnable features more uniformly in the early epochs compared to SGD
- Based on this analysis, proposes a simple clustering-based upsampling strategy for reducing simplicity bias / excessive reliance on fast-learnable features. The results show that this improves in-distribution generalization of standard small-scale image classification tasks.

**Strengths:**

- Simple easy-to-implement method that uses SAM and upsampling to improve in-distribution generalization
- The method is well justified with theoretical analysis comparing SAM and SGD on a toy data distribution. This analysis indicates that SAM is less sensitive to simplicity bias.

**Weaknesses:**

- No baselines. There are several papers now that try to reduce simplicity bias in order to improve performance:
    - https://arxiv.org/abs/2105.05612
    - https://arxiv.org/abs/2301.13293
    - https://arxiv.org/abs/2107.09044 (does not focus on simplicity bias explicitly, but similar to method proposed in paper)
    - simpler baselines: there are several papers that propose “example difficulty” metrics (https://arxiv.org/abs/2106.09647). How well do this correlate with the clusters found in your method? If you just train on the k examples with the highest difficulty scores (per class), does this fare worse than the proposed method?
- Limited novelty due to findings in [64] (Sharpness-aware minimization enhances feature quality via balanced learning). This paper also shows that SAM improves feature diversity (on real datasets + backed up with analysis on a toy dataset) and improves performance on transfer-learning tasks.
- Lacking discussion about when this method would fail. I can imagine two scenarios where the method would not work:
    1. Most training examples have one or more slow-learnable features. In this case, the clustering approach would “remove” most of the points in the dataset, and train on very points for multiple epochs. This could result in overfitting and performance that is worse than training. There’s an implicit assumption that there is some sort of one-to-one relation between examples and features. In the case where all examples contain an “easy” (e.g. patch) and a “hard” feature (e.g. CIFAR), would this method improve performance over SGD?
    2. In noisy datasets, low-quality examples or mislabeled examples would require more time to learn, and this method would cluster them and train on them for longer. That is, it would group examples that are “high-quality” and hard-to-learn with “low-quality” points. In this case, would the proposed method improve performance over SGD?
- “SB of SGD has been long conjectured to be the reason for the superior generalization performance of overparameterized models, by providing capacity control or implicit regularization” This incorrectly cites https://arxiv.org/abs/2006.07710v2, which shows that too much simplicity bias can lead to robustness and in-distribution generalization issues.
- Unfair evaluation. The experiments compare SAM+TA augmentation and SAM+USEFUL+TA to SGD (no TA). I think there should be two plots, complaring {SGD, SAM, SAM+USEFUL} w/ and w/o TA.
- Experiments on larger datasets. The image classification used here are fairly small-scale. I would like to how well this method scales to ImageNet-scale datasets (TinyImageNet is not a good proxy..)
- Writing is repetitive at times, especially the theory section (3.3)

**Questions:**

- How specific is the analysis to the toy distribution setting in which there is just 1 slow-learnable and 1 fast-learnable feature? How do things change if the number of slow-learnable features >> number of fast-learnable features?
- Why is max alignment between weight vector and ground-truth feature v_e the right way to evaluate the feature’s contribution to the model? Isn’t it hypothetically possible that SAM solutions rely more on the simpler feature if more weight vectors rely on v_e instead of v_d, even when max-alignment with slow feature is higher for SAM? Some discussion connecting this metric to “feature reliance” vis-a-vis model outputs would be great.

**Limitations:**

Please see strengths and weaknesses.

---

> ### Author Rebuttal · Authors · 2024-08-07
>
> Thanks for your feedback!
> 1. Comparison with simplicity bias baselines
> - Prior work, including papers referred to by the reviewer, showed the benefits of reducing the simplicity bias to **out-of-distribution (OOD), where there is a shift between training and test distribution** (c.f. [general comment](https://openreview.net/forum?id=yySpldUsU2&noteId=nTUcV9AnO0) for detailed discussion).
> - Our main contribution is to show that **reducing the simplicity bias benefits ID**. USEFUL is directly motivated by our Theorem 3, and we don’t see alternative methods for reducing the simplicity bias as *baselines* for our contribution. We conducted new experiments on CIFAR10 with EIIL & JTT. Our theoretically-motivated method outperforms EIIL and JTT in the ID setting (Table 5 in the **PDF**).
> 2. Example difficulty metrics vs USEFUL clusters & training on the k examples with highest difficulty scores per class
> - Methods to calculate example difficulty require either training the model to convergence (forgetting score, Toneva et al, NeurIPS’18) or training several models partially (El2N, Paul et al, NeurIPS’22). USEFUL only requires training one model for as few as 4 epochs. The fast-learnable cluster that we find is correlated with the easy examples, as we confirm in our **PDF**. But, USEFUL does not need a fine grained calculation of difficulty for all examples. In the method suggested by the reviewer, the optimal choice of K may be different in each class (since the learning difficulty of classes are often different), and finding the best K per class requires extensive hyperparameter tuning. USEFUL automatically finds the optimal choice for K in each class, and is cheap to apply.
> 3. Novelty w.r.t [64]
> - As discussed in our [general comment](https://openreview.net/forum?id=yySpldUsU2&noteId=nTUcV9AnO0), **prior work including [64] studied OOD**. [64] showed that SAM suppresses the spurious/redundant (**non-predictive**) features, and enables learning other/more diverse features which can benefit OOD. They studied toy/real **dataset with a known spurious feature and group labels**.
> - In contrast, we consider the **in-distribution (ID) setting and prove that SAM learns the same set of predictive features at a more uniform speed, which benefits ID**. Unlike [64], we do not require any group labels. To our knowledge, **our study is the first to show the benefits of reduced simplicity bias to ID, hence is pretty novel**.
> 4. Would USEFUL fail if:
> - Most training examples have one or more slow-learnable features
>   - USEFUL finds examples with **at least one fast-learnable feature** (and arbitrary number of slow-learnable features), and upsample the remaining examples. If most examples have at least one slow-learnable feature, nothing can be concluded. If there is no example with fast-learnable features, there is no simplicity bias to alleviate and there won't be any cluster with low training loss. But, this is unlikely for real-world datasets.
>   - Implicit assumption of one-to-one relation between examples and features. What if all examples contain an easy and a hard feature (e.g. CIFAR)? For simplicity of theoretical analysis, we assumed that every example contains one slow-learnable and at most one fast-learnable. But, **such assumption is not required in practice**, as we confirmed empirically on several benchmark datasets containing multiple slow and fast learnable features, including CIFAR10/100, TinyImageNet and CINIC10 where there is no one-to-one relation between examples and features.
> - Noisy labeled data. Our method and analysis consider a clean dataset. But, as we confirmed in our new experiments in Table 4 in the **PDF**, USEFUL can easily stack with robust methods for learning against noisy labels to reduce the simplicity bias and improve their performance.
> 5. Comparing {SGD, SAM, SAM+USEFUL} w/ and w/o TA
> - We compared (i) SGD + USEFUL with SAM, and (ii) SGD, SAM, SAM + TA when they stack with USEFUL to show that **USEFUL can stack with different gradient-based optimizers and data augmentations**. Our goal is not to compare methods w/ and w/o TA. Our evaluations are fair and confirm the effectiveness of USEFUL.
> 6. Larger datasets
> - Please see Q5 of [Reviewer Y69v](https://openreview.net/forum?id=yySpldUsU2&noteId=xu392aAv28). We do not believe there is anything specific to ImageNet that does not work with our method.
>
> **Questions**:
> 1. If #slow-learnable features >> #number of fast-learnable features
> - In this case, the dataset is very difficult to learn and learning is not affected much by the simplicity bias. In the extreme case where there is no fast learnable feature in the data, there won't be a low-loss cluster early in training and our method is not applicable (for the right reason). In practice, however, most datasets have several fast-learnable features and our method effectively improves the performance as we confirmed on several benchmark datasets.
> 2. Why is max alignment between weight vector and ground-truth feature v_e the right way to evaluate the feature’s contribution?
> - Since the model output of example $x_i$ at iteration t is given by $f(x_i; W) = \sum_{j \in [J]} ( y \beta_d^3 \langle w_j^{(t)}, v_d \rangle^3 + y \beta_e^3 \langle w_j^{(t)}, v_e \rangle^3 + \langle w_j^{(t)}, \xi_i \rangle^3 )$, the term $\beta_e^3 \max_{j \in [J]} \langle w_j^{(t)}, v_e \rangle^3$, i.e., max alignment between weight vector and the ground-truth feature $v_e$, greatly affects the prediction when the fast-learnable feature is present.
> 3. Isn’t it possible that SAM solutions rely more on the simpler feature if more weight vectors rely on v_e instead of v_d?
> - Indeed, SAM relies more on $v_e$ than $v_d$ in early training as proved in our Theorem 3.3. We showed that (1) SAM learns features **at a more uniform speed** as indicated by a smaller gap between features’ contribution in Theorem 3.4; and (2) SAM relies on $v_d$ **relatively** (w.r.t $v_e$) more than SGD in Theorem 3.5.

---

> ### Author Response · Authors · 2024-08-11
> **Your feedback is greatly appreciated**
>
> We hope our rebuttal has effectively addressed your concerns. As the discussion phase is nearing its conclusion, we are wondering if there's anything further we can clarify or elaborate on. Your feedback is greatly appreciated.

---

### Official Review · Reviewer_Y69v · 2024-07-20

**Soundness:** 4
**Presentation:** 3
**Contribution:** 3
**Rating:** 4
**Confidence:** 3

**Summary:**

This paper proposes an algorithm for changing the distribution of training data to improve the generalization of the model on origin data distribution. The paper is inspired by Sharpness Aware Minimization, which aims at finding a flat minimum meaning that it has a good generalization capability. This paper divides features into two categories: fast-learnable features and slow-learnable features and derives some observations like "SGD and SAM only learn fast-learnable or easy features early in training" and "SAM learns slow-learnable and fast-learnable features at a more uniform speed". The authors propose the method dubbed as USEFUL to train the model on some slow-learnable features repeatedly. The experiments show the effectiveness of USEFUL on CIFAR10 and CIFAR100 datasets.

**Strengths:**

- The paper is well-written and easy to follow.
- The paper has a theoretical analysis to analyze the learning progress and derive the proposed method.
- The experiments are abundant and comprehensive.

**Weaknesses:**

There are some questions based on the presentation of this paper, I will not hesitate to improve my score if the following question are solved.
- Difference between this paper and methods for long-tailed data distribution or measuring the difficulty of learning examples. Algorithms for long-tailed data distribution are usually based on resampling training data or reweighing loss value. The proposed USEFUL is similar to the resampling methods except that USEFUL focuses on the features that are hard/slow to learn. Some references for understanding: [Shi, Jiang-Xin, et al. "How re-sampling helps for long-tail learning?." Advances in Neural Information Processing Systems 36 (2023).](https://arxiv.org/pdf/2310.18236), [Shrivastava, Abhinav, Abhinav Gupta, and Ross Girshick. "Training region-based object detectors with online hard example mining." Proceedings of the IEEE conference on computer vision and pattern recognition. 2016.](https://arxiv.org/pdf/1604.03540v1) and some references based on it, [A Re-Balancing Strategy for Class-Imbalanced Classification Based on Instance Difficulty](https://openaccess.thecvf.com/content/CVPR2022/papers/Yu_A_Re-Balancing_Strategy_for_Class-Imbalanced_Classification_Based_on_Instance_Difficulty_CVPR_2022_paper.pdf), [Active Teacher for Semi-Supervised Object Detection](https://openaccess.thecvf.com/content/CVPR2022/papers/Mi_Active_Teacher_for_Semi-Supervised_Object_Detection_CVPR_2022_paper.pdf), I believe a discussion of these references in paper should be helpful.
- The relation between the proposed USEFUL and SAM? It seems like the motivation of USEFUL is changing the data distribution to get a flat minimum like SAM. But the results in Appendix D.2, *i.e.*, 53.8 for SGD 41.8 for SGD+USEFUL 12.4 for SAM in Table 1($\lambda_{max})$, do not show effectiveness compared with SAM. It could show the effectiveness on SGD but it's far from being comparable to SAM.
Some small questions:
- What's the exact formulation of the Data distribution?
- What's the "patch" meaning in Definition 3.1? Is that the same as the patch in ViT or the channel of the image? It's a little confusing.
- The experiments mainly focus on traditional architecture, e.g., n-Layer CNN, ResNet. More experiments on popular models and big datasets, e.g., Transformer ImageNet-1k, would be better.

**Questions:**

See Weakness.

**Limitations:**

N.A.

---

> ### Author Rebuttal · Authors · 2024-08-07
>
> We thank the reviewer for the valuable feedback and acknowledging our theoretical results and comprehensive experiments. We discuss the questions below.
>
> 1. **USEFUL vs resampling methods for long-tail data & example difficulty.**
> - As discussed in the [general comment](https://openreview.net/forum?id=yySpldUsU2&noteId=nTUcV9AnO0), long-tail data is an instance of OOD. Long-tail methods resample the data at the class or subclass level to **match the training and test distribution**. For example, [A Re-Balancing Strategy, CVPR’22] identifies instances with slow learning speed as more difficult instances and **dynamically increases their weights during training** to effectively change the data distribution to match the test distribution [page 2]. In contrast, we showed in the ID setting that the **simplicity bias of gradient-based methods makes them find solutions with suboptimal performance**. USEFUL **reduces the simplicity bias** by finding examples that are not learned during the first few training epochs, upsamples the rest of examples once, and restart training. Hence, USEFUL does not require calculating the difficulty of examples or do dynamic instance-wise reweighting or resampling during the training. Hence, the **effectiveness of our method is attributed to learning features at a more uniform speed, and not matching the training and test data distributions.** To our knowledge, the benefit of reduced simplicity bias to ID is studied for the first time by our work.
> - We conducted new experiments on long-tail CIFAR10 with an imbalance ratio of 10. Table 3 in the [PDF](https://openreview.net/forum?id=yySpldUsU2&noteId=nTUcV9AnO0) shows that our method can also improve the performance of SGD and SAM, by reducing the simplicity bias on the long-tail data. Fig 1 in the [PDF](https://openreview.net/forum?id=yySpldUsU2&noteId=nTUcV9AnO0) shows the distribution of classes before and after upsampling by USEFUL. Interestingly, we see that USEFUL upsamples more examples from some of the larger classes and still improves the accuracy on the balanced test set. This improvement is attributed to the more uniform speed of feature learning, and not balancing the training data distribution. Notably, *USEFUL outperforms class balancing to address long tail distribution*. Besides, USEFUL can be stacked with methods to address long tail data to further improve the performance, as we confirmed in our new experiments (please refer to Q3 of [Reviewer F4yx](https://openreview.net/forum?id=yySpldUsU2&noteId=3xcoRIZ69p)).
>
> 2. USEFUL’s solution is not as flat as SAM.
> - **The motivation of USEFUL is not to get a flatter minima like SAM**. In fact, **a flatter solution has been only conjectured (not proven) to have a better performance**, and recent studies have shown conflicting evidence on the relationship between flatness and generalization, suggesting that flatness does not fully explain SAM’s success [16 in paper, When do flat minima optimizers work?, NeurIPS’22, A modern look at the relationship between sharpness and generalization, ICML’23]. A key contribution of our work is identifying ***an orthogonal effect of SAM that is beneficial** **in-distribution (ID)**:* **we proved that SAM has less simplicity bias early in training and thus learns features at a more uniform speed.**
> - Our method, USEFUL, is **inspired by SAM in reducing the simplicity bias to improve the ID generalization performance**. Due to the same reason, **USEFUL can effectively improve the SAM’s performance itself**, as we confirmed empirically. While our goal is not to directly find a flatter minima or the same solution as SAM, we showed that USEFUL can find flatter minima. Note that to capture the sharpness/flatness, multiple different criteria have been proposed (largest Hessian eigenvalue and bulk of Hessian), and one criteria is not enough to accurately capture the sharpness. Table 1 shows that while the solution of SGD + USEFUL has a higher largest Hessian eigenvalue, it achieves the smallest bulk. Notably, the solution found by both SGD + USEFUL and SAM have lower sharpness in both indicators than SGD, proving that USEFUL successfully reduces the sharpness of the solution.
> 3. Formulation of the **Data distribution**.
> - The data distribution is defined in Definition 3.1 in Section 3.1. Each data point consists of 3 different patches, a slow-learnable, a fast-learnable, and random noise. Our data model is consistent with many prior theoretical works [2, 7, 10, 11, 15, 29, 37 in paper].
> 4. **"patch" meaning in Definition 3.1**.
> - It is similar to the patch in ViT. Each patch is a small region in the image and each image consists of P patches.
> 5. More experiments on popular models and big datasets, e.g., **Transformer** **ImageNet-1k**
> - Both CINIC10 and Tiny ImageNet contain images from ImageNet datasets and have a relatively large number of examples. CINIC10 dataset consists of 270,000 images, which is 4.5 times larger than CIFAR10. Tiny-ImageNet comprises 100,000 images distributed across 200 classes of ImageNet. Due to the computational constraints, we cannot conduct experiments on ImageNet (on our NVIDIA RTX A5000, a single training of ResNet18 with SGD on ImageNet takes around 1068 GPU hours). In Fig 5, we confirmed the effectiveness of USEFUL on other architectures such as ViT-small, which is a popular architecture for computer vision tasks [69 in paper, When vision transformers outperform resnets without pre-training or strong data augmentations, ICLR’22].

---

> ### Author Response · Authors · 2024-08-11
> **Your feedback is greatly appreciated**
>
> We hope our rebuttal has effectively addressed your concerns. As the discussion phase is nearing its conclusion, we are wondering if there's anything further we can clarify or elaborate on. Your feedback is greatly appreciated.

---

### Author Rebuttal · Authors · 2024-08-07

We thank the reviewers for their valuable feedback and recognizing the originality of our work. We’d like to first briefly emphasize the scope and contribution of our work:

- Our work shows, for the first time, that reducing the simplicity bias benefits **in-distribution (ID)**. Previously, the benefits of reducing simplicity bias have been only shown to **out-of-distribution (OOD)**, in particular to mitigate spurious correlations.
- **Prior methods: OOD setting** [long-tail data and spurious correlations]. Here, there is a shift between the training and test data distributions:
    - **Long-tail data** [Long-tailed CIFAR10, Inaturalist, ImageNetLT]: (sub)classes are highly imbalanced in training but are (more) balanced in test data (distribution shift). Upsampling the minority (sub)classes in training data improves the test performance on corresponding groups that are (much) larger in test data. Here, **the benefit comes from balancing the training data and matching the training and test data distributions**.
    - **Spurious features** [Waterbirds, CelebA, CMNIST, CIFAR-MNIST]: spurious features are simple features with a high correlation with a label at training time but not at test time (distribution shift). Due to simplicity bias, spurious features are learned *instead* of the more complex predictive features. This yields a poor (worst-group) accuracy on examples without the spurious feature at test time. Reducing the simplicity bias (or training with SAM) suppresses learning spurious features. Here, **the benefit comes from mitigating the spurious feature to learn more/other features**.
- **Our method: ID setting** [CIFAR10-100, TinyImageNet, CINIC10]**.** Here, training and test data have the same distribution, there is no spurious feature in the training data, and all features are predictive on the test set. Our work shows for the first time that reducing the simplicity bias benefits ID. Our contribution is orthogonal to the above work: **we proved that reducing the simplicity bias allows learning the *predictive* features *at a more uniform speed*, and showed that this benefits ID.** Unlike the OOD setting, *the benefit of reducing the simplicity bias in the ID setting is not attributed to balancing the training data distribution or suppressing the spurious features to learn other features*. It is attributed to the **(more uniform) speed/time of learning the same set of predictive features**. Our findings are more unexpected than its counterpart in the OOD setting and opens up a new direction for future research.
- **Our method: OOD setting** [Long-tailed CIFAR10, Waterbirds]. While our theory and main contribution is to show the benefits of reduced simplicity bias to ID, our new experiments show that our method can also benefit the long-tail setting, by reducing the simplicity bias and learning features of a class at a more uniform speed. Benefits of reducing the simplicity bias to the spurious setting has been studied by several prior work. While our experiments on Waterbirds shows that our method is also applicable to this setting, we note that **this is not our main contribution**.

We provide new experimental results on example difficulty, long-tail data, noisy labels, and spurious baselines in the attached **PDF**.

---

### Decision · Program_Chairs · 2024-09-25

**Decision:**

Accept (poster)

**Comment:**

This paper proposes a theoretically inspired method for modifying the distribution of the training data to improve generalization performance. Rather than building on generalization bounds, which are often too loose in practice to be useful, the paper utilizes SAM, a recent training scheme that has been empirically shown to be very effective for improving generalization, to drive its theoretician findings. Specifically, by analyzing the training dynamics of SAM and comparing that against SGD, the paper proves that SAM tends to learn hard and easy features at a more even pace, compared to SGD which heavily relies on easy features for training. Based on this theory, the authors hypothesize that the success of SAM may stem from its even-paced learning of both types of features. Inspired by this, they propose an algorithm for reweighting samples during training. The idea is to first train a model for a few epochs to get a rough initial model, and then use that model to cluster the data from each class into two categories, where one of them contains only samples that do not have any easy features. Then, by upsampling data that do not have easy features, the rest of the training balances the pace of learning easy and hard features. The proposed method is empirically verified on a few small but standard benchmark datasets, and the results confirm that the proposed reweighting method can indeed improve generalization performance when compared against SGD.

The paper has received a borderline score. Reviewers find the paper well-written and the theoretical analysis of SAM/SGD training dynamics, which demonstrates their connection to the speed at which they learn easy/hard features, to be original and intriguing. The majority of reviewers are also satisfied with the level of empirical verification of the algorithm, though they would appreciate it if the empirical analysis had covered larger datasets.

Common concerns among reviewers were the lack of novelty of the reweighting algorithm itself, as well as missing literature associated with data reweighting. The authors clarified that most of this literature relates to out-of-distribution (OOD) scenarios, while the paper focuses on in-distribution generalization. Other questions/concerns were about how the proposed algorithm works in the presence of label noise, and the effect of the proposed algorithm on the sharpness of the solution. The authors provided a detailed response to these concerns.

Unfortunately, two reviewers who gave the lowest initial rating (borderline reject) to this submission did not engage in the rebuttal, which makes it difficult to determine the fate of the paper. However, after carefully reading the authors' responses to these reviewers, I believe the provided responses can clarify some of the raised concerns. Furthermore, in agreement with the reviewers, I find the theoretical connection between SAM/SGD dynamics and the speed of easy/hard feature learning very intriguing, and I also find the proposed algorithm based on this analysis very simple to implement. Considering all the reviews and responses, I believe the contributions made by this work and the clarity of its presentation warrant an accept.